# LeakGFN: Robust Molecular Generation in Generative Flow Networks via Flow Decomposition

Hwanhee Kim [1]   Seungyeon Choi [1]   Sanghyun Park [1]

## Abstract

Generative Flow Networks (GFlowNets) have emerged as a powerful framework for molecular generation, sampling diverse candidates proportionally to a reward function. However, the vast chemical space necessitates truncating trajectory length, forcing models to treat incomplete molecular fragments as terminal states alongside valid molecules. This conflation distorts the learned distribution by allocating probability mass to chemically meaningless states. We propose **LEAKGFN**, a dual-head architecture that decomposes flow into two components: a *chemical head* modeling flow over the full chemical space, and a *valid head* estimating the fraction of flow reaching valid molecules within the truncation boundary. Through this decomposition, the valid head implicitly learns molecular reachability without explicit supervision. We prove that LEAKGFN recovers the correct distribution over accessible molecules under mild assumptions. Experiments on five molecular optimization tasks demonstrate consistent improvements over flow matching baselines, achieving state-of-the-art performance on four out of five tasks. Our plug-and-play module improves existing frameworks on pocket-conditioned and multi-objective tasks.

## 1. Introduction

Discovering novel molecules with desired properties is a fundamental challenge in drug discovery and material design (Hains et al., 2010; Hughes et al., 2011). While traditional virtual screening evaluates molecules from predefined libraries, *de novo* molecular generation methods explore a vast chemical space, estimated to contain $10^{23}$-

$10^{180}$ (Polishchuk et al., 2013), by generating structures from scratch. Recent deep generative models, including variational autoencoders (Gómez-Bombarelli et al., 2018; Jin et al., 2018), reinforcement learning (Olivecrona et al., 2017), and Generative Flow Networks (GFLOWNETs) (Bengio et al., 2021; 2023), have shown remarkable promise in this domain. GFLOWNETs are particularly attractive for molecular generation as they sample diverse candidates with probability proportional to a reward function (Bengio et al., 2021). Unlike reinforcement learning methods that converge to single optima, GFLOWNETs naturally explore multiple high-reward modes. This diversity is crucial for drug discovery, where multiple candidates are essential for downstream optimization (Jain et al., 2023; Zhu et al., 2023).

While GFLOWNETs effectively address the mode collapse problem in well-defined finite spaces, they face fundamental challenges when applied to domains with immense state spaces, such as chemical space. To make computation tractable in such vast spaces, existing works typically truncate the trajectory length, thereby constraining the reachable state space to a manageable size. Although this truncation is necessary for computational efficiency, it introduces a critical *target distribution mismatch*: while flow conservation still holds within the truncated DAG, the model learns a distribution over the incorrect target space that includes both valid molecules and incomplete fragments. As illustrated in Figure 1(A), conventional approaches treat states at the truncation boundary as *forced terminals*, assigning them probability mass alongside valid molecules. This conflates two fundamentally different outcomes that complete molecules with meaningful rewards, and incomplete fragments that happen to reach the length limit. Consequently, the learned distribution allocates probability to chemically meaningless states, reducing the effective capacity for exploring valid chemical space.

Prior works have addressed related challenges through improved training objectives (Madan et al., 2023; Pan et al., 2023), exploration strategies (Kim et al., 2024b;a), and active learning (Jain et al., 2022). However, none directly tackle the fundamental issue of learning over an incorrect target space induced by truncation. This problem becomes increasingly severe for larger molecules, which is precisely

[1]Department of Computer Science, Yonsei University, Seoul, Korea. Correspondence to: Sanghyun Park <sanghyun@yonsei.ac.kr>.

*Proceedings of the 43rd International Conference on Machine Learning*, Seoul, South Korea. PMLR 306, 2026. Copyright 2026 by the author(s).

the regime relevant for complex drug scaffolds.

We propose LEAKGFN, which learns the correct distribution over valid molecules by explicitly decomposing flow into two components (Figure 1(B)). The *chemical flow $F_{\text{chem}}$* represents flow over the entire chemical space without truncation constraints, while the *valid flow $F_{\text{valid}}$* represents flow toward molecules that complete within the truncation boundary. Our architecture employs a *chemical head* that models flow magnitude over all transitions, and a *valid head* that estimates the fraction of flow reaching accessible terminals. Through the decomposition $F_{\text{valid}} = F_{\text{chem}} \cdot \lambda$, the valid head implicitly learns to predict reachability without explicit supervision. This allows LEAKGFN to learn as if operating in an untruncated space restricted to accessible molecules. We prove that this decomposition recovers the correct distribution over valid, reachable molecules under mild assumptions.

Crucially, our dual-head decomposition improves *training dynamics* near the truncation boundary rather than merely changing the target distribution. By providing the model with separate gradient pathways for chemical flow and reachability estimation, the architecture resolves conflicting learning signals that arise when a single flow function must simultaneously model flow toward valid terminals and flow toward the boundary. Our ablation studies confirm that this training dynamics improvement—not the exploration strategy—accounts for the dominant performance gain.

Our main contributions are as follows:

- **Problem Identification and Formalization.** We identify that trajectory truncation causes GFlowNets to learn over an incorrect target space that conflates valid molecules with incomplete fragments, and provide the first formal definition of this *flow leakage* phenomenon (Definition 3.1).

- **LeakGFN: A Principled Solution.** We propose a dual-head architecture that decomposes flow into chemical and valid components, improving training dynamics by decoupling gradient signals for flow magnitude and reachability. The valid head implicitly learns molecular reachability through asymmetric terminal treatment, requiring no explicit supervision. We prove that this decomposition recovers the correct distribution over accessible molecules.

- **Comprehensive Empirical Validation.** We demonstrate state-of-the-art performance on four of five molecular optimization benchmarks, with notable robustness to hyperparameter selection ($L_{\max}$). Ablation studies confirm the decomposition accounts for the dominant improvement independent of exploration strategy. Our plug-and-play module improves existing

frameworks on pocket-conditioned and multi-objective generation tasks.

## 2. Related Work

**Generative Flow Networks.** GFlowNets (Bengio et al., 2021) sample compositional objects proportionally to a reward function. Various training objectives have been proposed: flow matching (FM) (Bengio et al., 2021), trajectory balance (TB) (Malkin et al., 2022), detailed balance (Bengio et al., 2023), and subtrajectory balance (Madan et al., 2023). Our method is orthogonal to these objectives. For molecular generation, both atom-level (Bengio et al., 2021) and fragment-based (Jin et al., 2020; Bengio et al., 2023) approaches have been explored, with recent extensions to pocket-conditioned generation (Shen et al., 2024; Jiang et al., 2024). Our dual-head architecture integrates seamlessly into these frameworks.

**Exploration and Large State Spaces.** Effective exploration strategies include $\epsilon$-greedy, Thompson sampling (Rector-Brooks et al., 2023), local search (Kim et al., 2024b), and genetic operators (Kim et al., 2024a). For large state spaces, Pan et al. (2023) proposed forward-looking objectives. However, none of these methods address the fundamental issue that truncation causes learning over an incorrect target space. To our knowledge, LEAKGFN is the first to explicitly model flow leakage and provide theoretical guarantees for the correct distribution over accessible molecules.

## 3. Method

We first review GFlowNets and formally characterize the truncation problem, then present LEAKGFN.

### 3.1. Preliminaries: GFlowNets

A Generative Flow Network (GFlowNet) (Bengio et al., 2021) learns to sample compositional objects $x \in \mathcal{X}$ with probability proportional to a reward function $R(x) \geq 0$. Objects are constructed through a sequence of actions in a directed acyclic graph (DAG) $\mathcal{G} = (\mathcal{S}, \mathcal{A})$, where $\mathcal{S}$ denotes states and $\mathcal{A}$ denotes actions (edges). Each trajectory $\tau = (s_0 \to s_1 \to \cdots \to s_n)$ starts from an initial state $s_0$ and terminates at a terminal state $s_n \in \mathcal{X}$.

GFlowNets define a *flow function* $F : \mathcal{A} \to \mathbb{R}_{\geq 0}$ over edges, where $F(s \to s')$ denotes the flow along the edge from state $s$ to state $s'$. The key principle is *flow conservation*, which requires that for any non-terminal state $s$, the incoming flow equals the outgoing flow:

$$\sum_{s' \in \text{Pa}(s)} F(s' \to s) = \sum_{s'' \in \text{Ch}(s)} F(s \to s'') \qquad (1)$$

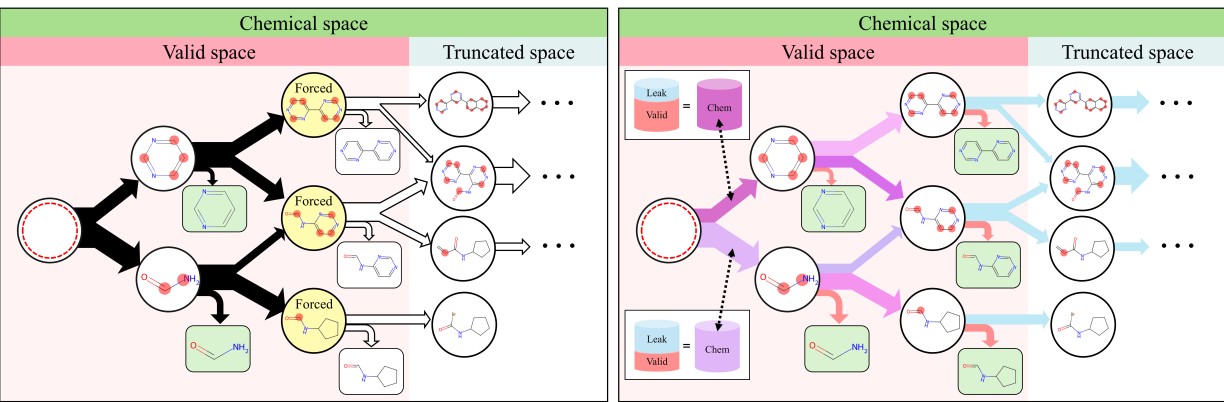

*Figure 1.* Comparison of state space formulations under trajectory truncation. **(A) Forced Termination:** Conventional GFlowNets treat states at the truncation boundary as forced terminals (yellow), assigning them probability mass alongside valid terminals (green). Flow beyond the boundary (white) is ignored, resulting in a distribution over both valid molecules and incomplete structures. **(B) Flow Decomposition (LeakGFN):** Our method decomposes flow into valid (red) and leak (blue) components. The chemical flow (purple) at each state is a mixture that gradually separates as trajectories progress. Only valid flow contributes to the learned distribution, while leak flow explicitly models probability mass that would exit the truncation boundary. This decomposition enables learning the correct distribution exclusively over valid, accessible molecules.

where $\mathrm{Pa}(s)$ and $\mathrm{Ch}(s)$ denote parents and children of $s$, respectively. For terminal states $x \in \mathcal{X}$, the incoming flow equals the reward:

$$\sum_{s' \in \mathrm{Pa}(x)} F(s' \to x) = R(x) \qquad (2)$$

When these conditions are satisfied, sampling trajectories by following the forward policy $\pi(s'|s) \propto F(s \to s')$ yields terminal states with the desired distribution:

$$P(x) = \frac{R(x)}{\sum_{x' \in \mathcal{X}} R(x')} \propto R(x) \qquad (3)$$

### 3.2. Problem: Truncation Induces Incorrect Target Space

In molecular generation, the state space is enormous. To ensure computational tractability, existing methods impose a maximum trajectory length $L_{\max}$, truncating generation at a fixed depth (Bengio et al., 2023). This creates two disjoint types of terminal states ($\mathcal{T}_{\mathrm{valid}} \cap \mathcal{T}_{\mathrm{forced}} = \emptyset$). *Valid terminals* $\mathcal{T}_{\mathrm{valid}}$ represent states where the agent *voluntarily* chose the stop action, producing a complete molecule. *Forced terminals* $\mathcal{T}_{\mathrm{forced}}$ represent states that reach $L_{\max}$ without the agent choosing to stop—termination is imposed by the environment rather than the agent's policy.

Conventional GFlowNets treat both as terminals, learning a distribution over $\mathcal{T}_{\mathrm{valid}} \cup \mathcal{T}_{\mathrm{forced}}$. However, this is fundamentally incorrect. Forced terminals are incomplete fragments, not valid molecules. We formally characterize this phenomenon:

**Definition 3.1** (Flow Leakage). Given a GFlowNet with truncation boundary $L_{\max}$, *flow leakage* is the probability mass allocated to forced terminal states $\mathcal{T}_{\mathrm{forced}}$ under the learned distribution:

$$\mathcal{L}_{\mathrm{leak}} = \sum_{s \in \mathcal{T}_{\mathrm{forced}}} P(s) = \frac{\sum_{s \in \mathcal{T}_{\mathrm{forced}}} R(s)}{\sum_{s \in \mathcal{T}_{\mathrm{forced}}} R(s) + \sum_{x \in \mathcal{T}_{\mathrm{acc}}} R(x)} \qquad (4)$$

where $P(s)$ is the probability assigned to state $s$ under the converged GFlowNet policy. Flow leakage directly reduces the effective probability mass available for sampling valid molecules, distorting the learned distribution away from the target.

The correct target is the distribution over *accessible* valid molecules:

$$P_{\mathrm{target}}(x) = \frac{R(x)}{\sum_{x' \in \mathcal{T}_{\mathrm{acc}}} R(x')} \quad \text{for } x \in \mathcal{T}_{\mathrm{acc}} \qquad (5)$$

where $\mathcal{T}_{\mathrm{acc}}$ denotes the set of valid molecules reachable within $L_{\max}$ steps, $R(x)$ is the reward for molecule $x$, and the denominator normalizes over all accessible molecules.

### 3.3. LeakGFN: Flow Decomposition

Our key insight is that flow at any state can be decomposed into two components based on reachability within the truncation boundary.

**Dual-Head Architecture.** We propose a dual-head architecture that shares a common state-action encoder $\theta_{\mathrm{emb}}$ but

uses separate heads for predicting flow magnitude and reachability.

The *chemical head* $\theta_{\text{chem}}$ models flow over the entire chemical space without artificial truncation constraints:

$$F_{\text{chem}}(s, a) = \exp\big(\theta_{\text{chem}}(\theta_{\text{emb}}(s, a))\big) \qquad (6)$$

where $\theta_{\text{emb}}(s, a)$ is a shared embedding of the state-action pair, and the exponential ensures non-negative flow. This head has the same architecture as standard GFlowNet flow predictors, outputting log-scale flow values.

The *valid head* $\theta_{\text{valid}}$ estimates the fraction of flow that will reach accessible terminal states within the truncation boundary:

$$\lambda(s, a) = \sigma\big(\theta_{\text{valid}}(\theta_{\text{emb}}(s, a))\big) \qquad (7)$$

where $\sigma(\cdot)$ denotes the sigmoid function, ensuring $\lambda(s, a) \in [0, 1]$. Intuitively, $\lambda(s, a)$ represents the fraction of flow through action $a$ at state $s$ that will terminate at a valid molecule rather than being truncated.

The *valid flow*, used exclusively for sampling, combines these components for add actions:

$$F_{\text{valid}}(s, a) = F_{\text{chem}}(s, a) \cdot \lambda(s, a) \quad \text{for } a \in \mathcal{A}_{\text{add}} \qquad (8)$$

where $\mathcal{A}_{\text{add}}$ denotes the set of fragment addition actions. The complementary *leak flow* $F_{\text{leak}}(s, a) = F_{\text{chem}}(s, a) \cdot (1 - \lambda(s, a))$ represents flow that would exit through the truncation boundary. For the stop action, we use a single shared head $F(s, \text{stop})$ without decomposition. This design choice reflects that the stop action directly terminates generation at the current state with reward $R(s)$, requiring no reachability estimation—the molecule is already complete when stop is selected. As illustrated in Figure 1(B), the chemical flow (purple) at early states is a mixture that gradually separates into valid flow (red) toward accessible molecules and leak flow (blue) toward the boundary.

### 3.4. Training Objective

The cornerstone of LEAKGFN's implicit reachability learning is the asymmetric treatment of forced terminals between the two heads, which enables $\lambda$ to capture molecular accessibility without explicit supervision.

**Asymmetric Terminal Treatment.** We train the two heads with different treatments of terminal states, summarized in Table 1.

For valid terminals $s \in \mathcal{T}_{\text{valid}}$, both heads apply the standard terminal condition:

$$F(s, \text{stop}) = R(s) \qquad (9)$$

*Table 1.* Terminal treatment for each head during training.

| | $F_{\text{chem}}$ | $F_{\text{valid}}$ |
|---|---|---|
| Valid terminal | Terminal | Terminal |
| Forced terminal | Non-terminal | Terminal |

For forced terminals $s \in \mathcal{T}_{\text{forced}}$, the treatments differ. The valid head treats them as terminals with their oracle rewards:

$$F_{\text{valid}}^{\text{in}}(s) = R(s) \qquad (10)$$

where $R(s)$ is the oracle evaluation of the incomplete fragment. In fragment-based generation, every state corresponds to a valid molecular graph that the oracle can evaluate. Forced terminals typically receive low rewards ($R(s) < 0.1$) because incomplete fragments with open attachment points lack the structural completeness required for high bioactivity or drug-likeness. The chemical head treats them as non-terminals, enforcing flow conservation:

$$F_{\text{chem}}^{\text{in}}(s) = F_{\text{chem}}^{\text{out}}(s) = F(s, \text{stop}) + \sum_{a \in \mathcal{A}_{\text{add}}} F_{\text{chem}}(s, a) \qquad (11)$$

This allows $F_{\text{chem}}$ to model outgoing flow through add actions as if the trajectory could continue beyond the truncation boundary.

**Implicit Learning of $\lambda$.** This asymmetric treatment induces action-level reachability learning without explicit supervision. At forced terminal $s$, $F_{\text{valid}}$ computes both terminal loss and flow conservation loss:

$$F_{\text{valid}}^{\text{in}}(s) = R(s) \qquad \text{(terminal condition)} \qquad (12)$$
$$F_{\text{valid}}^{\text{in}}(s) = F_{\text{valid}}^{\text{out}}(s) \qquad \text{(flow conservation)} \qquad (13)$$

Thus $F_{\text{valid}}^{\text{out}}(s) = F(s, \text{stop}) + \sum_a F_{\text{valid}}(s, a) = R(s)$.

A key challenge is ensuring $F(s, \text{stop}) = R(s)$ at terminals. Without explicit supervision, the stop action logit may diverge from the true reward. To address this, we add an auxiliary self-loop transition $s \xrightarrow{\text{stop}} s$ for every terminal state (both valid and forced), treated as a valid terminal with reward $R(s)$. Concretely, for each terminal state $s$ encountered during training, we append a self-loop transition $(s, \text{stop}, s, R(s))$ to the training batch. This explicitly trains $F(s, \text{stop}) = R(s)$ via the terminal loss. Combined with flow conservation, this forces:

$$\sum_{a \in \mathcal{A}_{\text{add}}} F_{\text{valid}}(s, a) = 0 \qquad (14)$$

Meanwhile, $F_{\text{chem}}$ computes only flow conservation loss at forced terminals, maintaining positive flow through add actions ($F_{\text{chem}}(s, a) > 0$ for $a \in \mathcal{A}_{\text{add}}$). This forces:

$$\lambda(s, a) = \frac{F_{\text{valid}}(s, a)}{F_{\text{chem}}(s, a)} = 0 \quad \text{for all } a \in \mathcal{A}_{\text{add}} \qquad (15)$$

This learning signal propagates backward through the DAG via the flow matching objective. At states along trajectories that reach valid terminals within the truncation boundary, $\lambda(s, a)$ remains close to 1, while states leading toward the boundary have $\lambda(s, a) \approx 0$. Thus $\lambda$ learns to predict reachability without explicit supervision.

**Flow Matching Loss.** We train both heads using the flow matching objective. For $F_{valid}$, we decompose the loss into terminal and flow conservation components:

$$\mathcal{L}_{valid}^{term} = \sum_{s \in \mathcal{T}_{valid} \cup \mathcal{T}_{forced}} \left( \log \frac{F_{valid}^{in}(s)}{R(s)} \right)^2 \quad (16)$$

$$\mathcal{L}_{valid}^{flow} = \sum_{s \notin \mathcal{T}_{valid}} \left( \log \frac{F_{valid}^{in}(s)}{F_{valid}^{out}(s)} \right)^2 \quad (17)$$

where the terminal loss (Eq. 16) is applied at all terminals (both valid and forced), and the flow conservation loss (Eq. 17) is applied at non-valid-terminal states (i.e., non-terminals and forced terminals).

For $F_{chem}$, we apply only the flow conservation loss at non-valid-terminal states:

$$\mathcal{L}_{chem} = \sum_{s \notin \mathcal{T}_{valid}} \left( \log \frac{F_{chem}^{in}(s)}{F_{chem}^{out}(s)} \right)^2 \quad (18)$$

The key difference from conventional GFlowNets lies in the treatment of forced terminals. In standard flow matching, forced terminals are treated purely as terminals—only terminal loss is applied, with no flow conservation constraint. In LEAKGFN, $F_{valid}$ computes *both* terminal loss and flow conservation loss at forced terminals, explicitly enforcing $\sum_a F_{valid}(s, a) = 0$ through gradient descent. Meanwhile, $F_{chem}$ computes only flow conservation loss at forced terminals, maintaining $F_{chem}(s, a) > 0$. This asymmetry enables the implicit learning of $\lambda(s, a) = 0$ at forced terminals.

The total loss is $\mathcal{L} = \mathcal{L}_{valid} + \alpha \mathcal{L}_{chem}$, where $\mathcal{L}_{valid} = \gamma \mathcal{L}_{valid}^{term} + \mathcal{L}_{valid}^{flow}$ with terminal loss coefficient $\gamma$, and $\alpha \geq 0$ controls the strength of chemical flow regularization. A larger $\alpha$ encourages the model to learn flow dynamics that extend beyond forced terminals, improving the estimation of leak flow at states near the truncation boundary. When $\alpha = 0$, the chemical head receives no direct supervision; however, it is still trained indirectly through the coupling $F_{valid} = F_{chem} \cdot \lambda$, as gradients from $\mathcal{L}_{valid}$ backpropagate through both heads. We use $\alpha = 1$ in all experiments unless otherwise specified. We provide sensitivity analysis on $\alpha$ in Appendix C.

## 3.5. Flow-Guided Exploration

Standard GFlowNet training uses $\epsilon$-exploration, mixing the learned policy with uniform random actions:

$$\pi_{explore}(a|s) = (1 - \epsilon) \cdot \pi(a|s) + \epsilon \cdot \text{Uniform}(a) \quad (19)$$

where $\pi(a|s)$ is the learned policy, $\epsilon \in [0, 1]$ controls the exploration rate, and Uniform$(a)$ assigns equal probability to all valid actions.

Under truncation, this strategy is inefficient. Uniform sampling wastes probability on actions leading toward the truncation boundary, generating trajectories that end at forced terminals and provide no learning signal.

We propose an alternative strategy where exploration actions are sampled proportionally to chemical flow:

$$\pi_{explore}(a|s) \propto F_{chem}(s, a) \quad (20)$$

**Comparison with $\epsilon$-exploration.** Standard $\epsilon$-exploration samples uniformly over all valid actions, including those leading toward truncation. In contrast, our flow-guided strategy (Eq. 20) naturally down-weights such actions since $F_{chem}$ captures the flow structure of the chemical space. This results in more sample-efficient training without requiring explicit action masking.

We use $F_{chem}$ rather than $F_{valid}$ for exploration because $F_{valid}$ diminishes near the truncation boundary as probability mass leaks into invalid states. In contrast, $F_{chem}$ remains well-defined throughout the chemical space, providing stable gradients and guiding exploration toward chemically plausible regions.

## 3.6. Theoretical Guarantee

We now show that LEAKGFN recovers the correct distribution over accessible molecules.

**Theorem 3.2** (Correctness of LeakGFN). *Suppose the flow matching conditions hold for $F_{valid}$: (i) flow conservation at non-terminal states, and (ii) terminal flow equals reward at accessible valid terminals. Formally,*

$$F_{valid}^{in}(s) = F_{valid}^{out}(s) \qquad \forall s \notin \mathcal{T}_{valid} \quad (21)$$

$$F_{valid}^{in}(x) = R(x) \qquad \forall x \in \mathcal{T}_{acc} \quad (22)$$

*Then sampling trajectories by following the forward policy $\pi(a|s) \propto F_{valid}(s \to s')$ yields the target distribution $P(x) = R(x)/Z_{valid}$ for all accessible valid terminals $x \in \mathcal{T}_{acc}$, where the partition function $Z_{valid} = \sum_{x' \in \mathcal{T}_{acc}} R(x')$ normalizes over all accessible molecules.*

*Proof.* Let $Z_{valid} = \sum_{s' \in \text{Ch}(s_0)} F_{valid}(s_0 \to s')$ be the total valid flow leaving the initial state $s_0$, where $\text{Ch}(s_0)$ denotes

the children of $s_0$. By condition (i), flow is conserved at all non-terminal states. Thus, total flow leaving $s_0$ equals total flow entering terminals:

$$Z_{\text{valid}} = \sum_{x \in \mathcal{T}_{\text{acc}}} \sum_{s'} F_{\text{valid}}(s' \to x) = \sum_{x \in \mathcal{T}_{\text{acc}}} R(x) \quad (23)$$

where the last equality follows from condition (ii). The sampling probability for terminal $x$ is:

$$P(x) = \frac{\sum_{s'} F_{\text{valid}}(s' \to x)}{Z_{\text{valid}}} = \frac{R(x)}{\sum_{x' \in \mathcal{T}_{\text{acc}}} R(x')} \quad (24)$$

$\square$

This theorem confirms that by training on $F_{\text{valid}}$ alone, LEAKGFN learns the target distribution (Equation 5) over accessible molecules, completely ignoring the corrupting influence of forced terminals.

*Remark* 3.3. Theorem 3.2 provides correctness conditional on $F_{\text{valid}}$ satisfying the flow-matching conditions at convergence. Like prior GFlowNet analyses (Bengio et al., 2021; Malkin et al., 2022), we do not provide formal convergence guarantees—this remains an open problem for flow-based generative models. We empirically verify that the learned model achieves the desired behavior through extensive experiments in Section 4.

## 4. Experiments

We evaluate LEAKGFN on a diverse set of molecular optimization tasks, comparing against state-of-the-art baselines and conducting comprehensive ablation studies.

### 4.1. Experimental Setup

**Tasks.** We evaluate on five optimization tasks spanning diverse molecular properties. The kinase inhibition tasks target JNK3, GSK3$\beta$, and DRD2, three therapeutically relevant targets where binding affinity is predicted using a random forest surrogate trained on bioactivity data. The molecular property tasks include QED (quantitative estimate of drug-likeness) and SA (synthetic accessibility).

**Baselines.** We compare against several categories of methods: (1) genetic algorithm baselines (Graph-GA) (Jensen, 2019); (2) reinforcement learning methods including REINVENT (Blaschke et al., 2020) and GEGL (Ahn et al., 2020); and (3) GFlowNet variants including the standard flow matching objective (GFN-FM) (Bengio et al., 2021), trajectory balance (GFN-TB) (Malkin et al., 2022), subtrajectory balance (GFN-SubTB) (Madan et al., 2023) and dynamic batching (GFN-DB) (Malkin et al., 2023).

**Metrics.** We evaluate methods using three complementary metrics with $k = 100$: (1) *Score*, the mean oracle score of the top-$k$ molecules; (2) *Diversity* (Div), the average pairwise Tanimoto dissimilarity (on Morgan fingerprints) among top-$k$ molecules; and (3) *Uniqueness* (Uni), the fraction of unique molecules among all 1,000 generated samples. We report the harmonic mean (HM) of these three metrics as the primary evaluation criterion. Unlike arithmetic mean, HM is sensitive to low values, ensuring that methods cannot achieve high scores by excelling in one metric while neglecting others. All experiments are conducted with 3 random seeds, and we report mean $\pm$ standard deviation.

**Implementation.** We use a fragment-based molecular representation where molecules are constructed by iteratively attaching fragments from a vocabulary of 105 building blocks. The maximum trajectory length is set to $L_{\max} = 8$ fragments unless otherwise specified. Both the chemical head and valid head use a 3-layer graph neural network with 256 hidden dimensions. Additional details are provided in Appendix A.

### 4.2. Single-Objective Molecular Optimization

Table 2 presents results across five molecular optimization tasks. Methods are grouped by approach: non-GFlowNet baselines (top), GFlowNet variants with alternative training objectives (middle), and flow matching based methods (bottom). LEAKGFN achieves the best performance on four out of five tasks and remains competitive on the remaining task. All main results in Table 2 use $\alpha = 1$.

**Comparison with Non-GFlowNet Methods.** All GFlowNet variants significantly outperform genetic algorithms (Graph-GA) and reinforcement learning methods (REINVENT, GEGL) on most tasks, confirming the advantage of diversity-seeking generation for molecular optimization. The performance gap is particularly large on drug-likeness metrics (QED, SA), where the ability to explore multiple modes is crucial.

**Comparison with GFN-FM.** Since LEAKGFN extends GFN-FM by adding the dual-head architecture, the bottom group of Table 2 enables direct comparison that isolates the contribution of flow decomposition. LEAKGFN consistently matches or outperforms GFN-FM across all five tasks, demonstrating that our approach provides a strictly beneficial inductive bias without introducing trade-offs.

The improvement is most pronounced on JNK3, where LEAKGFN achieves a 62% relative improvement in HM score from 0.403 to 0.653. Notably, GFN-FM exhibits high variance on this task with standard deviation of 0.218, indicating unstable training likely caused by the conflation of valid and forced terminals. By explicitly decomposing

*Table 2.* Comparison of molecular optimization methods on the diverse molecular properties. We report HM score($\uparrow$) averaged over 3 random seeds. Best results are in **bold**, second best are underlined.

|  | JNK3 | GSK3 | DRD2 | QED | SA |
|---|---|---|---|---|---|
| Graph-GA | $0.521 \pm 0.009$ | $0.558 \pm 0.006$ | $0.667 \pm 0.029$ | $0.709 \pm 0.008$ | $0.625 \pm 0.008$ |
| REINVENT | $0.560 \pm 0.071$ | $0.612 \pm 0.029$ | $0.534 \pm 0.059$ | $0.692 \pm 0.104$ | $0.540 \pm 0.006$ |
| GEGL | $0.593 \pm 0.039$ | $0.625 \pm 0.142$ | $0.710 \pm 0.071$ | $0.869 \pm 0.009$ | $0.674 \pm 0.009$ |
| GFN-DB | $0.623 \pm 0.003$ | $0.673 \pm 0.001$ | $\mathbf{0.934 \pm 0.002}$ | $0.897 \pm 0.044$ | $0.864 \pm 0.007$ |
| GFN-TB | $0.624 \pm 0.003$ | $0.675 \pm 0.008$ | $\underline{0.933 \pm 0.000}$ | $\underline{0.899 \pm 0.047}$ | $0.867 \pm 0.002$ |
| GFN-SubTB | $0.620 \pm 0.009$ | $0.686 \pm 0.015$ | $0.845 \pm 0.039$ | $0.845 \pm 0.007$ | $0.880 \pm 0.008$ |
| GFN-FM | $0.403 \pm 0.218$ | $\underline{0.716 \pm 0.071}$ | $0.775 \pm 0.041$ | $\mathbf{0.926 \pm 0.003}$ | $\underline{0.922 \pm 0.002}$ |
| **Ours (LeakGFN)** | $\mathbf{0.653 \pm 0.068}$ | $\mathbf{0.760 \pm 0.008}$ | $0.829 \pm 0.122$ | $\mathbf{0.926 \pm 0.002}$ | $\mathbf{0.923 \pm 0.003}$ |

flow, LEAKGFN reduces this variance by 69%, from 0.218 to 0.068. Similar variance reduction is observed on GSK3, where variance decreases by 89% from 0.071 to 0.008, confirming that flow decomposition stabilizes training across challenging optimization landscapes. Even on tasks where GFN-FM already performs well (QED, SA), LEAKGFN maintains comparable performance while achieving lower variance. Detailed per-metric breakdowns in Appendix B reveal that this improvement primarily stems from stabilizing Uniqueness: GFN-FM exhibits severe Uniqueness collapse on JNK3 ($0.444 \pm 0.468$), while LeakGFN recovers it to $0.930 \pm 0.063$.

**Comparison with Other GFlowNet Variants.** Among GFlowNet methods, LEAKGFN achieves the highest scores on JNK3, GSK3, QED, and SA. On DRD2, GFN-DB achieves the best performance of 0.934, followed closely by GFN-TB at 0.933, while LEAKGFN obtains 0.797. We attribute this gap to the task characteristics: DRD2's high-reward molecules tend to complete within shorter trajectories compared to kinase inhibition tasks, resulting in minimal flow leakage at $L_{\max} = 8$ where LeakGFN's benefits are limited. However, as we demonstrate in Table 3, the robustness advantage of LeakGFN becomes apparent when $L_{\max}$ is set conservatively larger.

**Robustness to $L_{\max}$ Selection.** A critical question for practitioners is how methods perform when the optimal trajectory length is unknown a priori. Table 3 examines this on DRD2 across all GFlowNet variants. At the well-tuned $L_{\max} = 8$, trajectory-level methods (GFN-DB, GFN-TB, GFN-SubTB) achieve excellent performance, with GFN-DB reaching 0.934. However, all trajectory-level methods exhibit *catastrophic sensitivity* to this hyperparameter: GFN-DB collapses from 0.934 at $L_{\max} = 8$ to 0.096 at $L_{\max} = 12$, GFN-TB from 0.933 to 0.106, and GFN-SubTB from 0.845 to 0.107. In contrast, LEAKGFN demonstrates *graceful degradation*, maintaining reasonable performance of 0.435 at $L_{\max} = 12$—over $4\times$ higher than all trajectory-

level baselines.

This robustness is crucial for real-world drug discovery, where the target molecule size distribution is often unknown and practitioners prefer conservative, larger $L_{\max}$ values to avoid excluding potentially valuable candidates. Trajectory-level objectives (TB, DB, SubTB) receive learning signal only at terminals, making them highly sensitive to the proportion of forced terminals which increases rapidly with $L_{\max}$. State-level methods (FM, LEAKGFN) provide direct supervision at each state, offering inherent robustness. LEAKGFN further improves upon GFN-FM by explicitly modeling flow leakage, achieving the best performance at $L_{\max} \geq 10$. We verify robustness to the hyperparameter $\alpha$ in Appendix C, finding consistent performance across $\alpha \in [0, 2]$.

*Table 3.* Robustness to $L_{\max}$ on DRD2. HM (mean) over 3 seeds.

| Method | 6 | 8 | 10 | 12 |
|---|---|---|---|---|
| Ours(LEAKGFN) | 0.865 | 0.797 | **0.517** | **0.435** |
| GFN-FM | 0.859 | 0.775 | 0.356 | 0.380 |
| GFN-DB | 0.926 | **0.934** | 0.136 | 0.096 |
| GFN-TB | 0.906 | 0.933 | 0.132 | 0.106 |
| GFN-SubTB | **0.927** | 0.845 | 0.158 | 0.107 |

### 4.3. Ablation Studies

We use JNK3 as the primary ablation task due to its challenging optimization landscape and sensitivity to truncation effects. All experiments use $L_{\max} = 8$ unless otherwise specified and are averaged over 3 random seeds. Table 4 presents ablation results removing key components of LEAKGFN. Removing flow-guided exploration (w/o $\pi_{\text{explore}}$) decreases performance from 0.64 to 0.62 in HM score, with notable drops in uniqueness from 0.86 to 0.81 and success rate from 0.64 to 0.51. This confirms that biasing exploration toward chemically meaningful transitions improves sample efficiency, though the contribution is modest ($\Delta$HM = 0.02).

Removing the valid head entirely, which is equivalent to setting $\lambda = 1$, causes dramatic performance degradation from 0.64 to 0.40 ($\Delta$HM $= 0.24$). Uniqueness drops from 0.86 to 0.44, indicating that without explicit flow decomposition, the model collapses to generating duplicate forced terminals. Comparing the two ablations, the dual-head decomposition accounts for the dominant improvement ($+0.22$ HM) independent of exploration strategy ($+0.02$ HM). This confirms that the architecture's primary contribution is improving *training dynamics* near the truncation boundary—by decoupling flow magnitude from reachability estimation, the model avoids conflicting gradients that destabilize single-head architectures.

*Table 4.* Ablation study on JNK3. All metrics are averaged over 3 seeds. Success denotes the fraction of top-100 molecules with score $\geq 0.5$.

| Method | HM | Score | Div | Uni | Success |
|---|---|---|---|---|---|
| LEAKGFN (full) | **0.64** | **0.55** | 0.60 | **0.86** | **0.64** |
| w/o $\pi_{\text{explore}}$ | 0.62 | 0.51 | **0.64** | 0.81 | 0.51 |
| w/o $\theta_{valid}$ ($\lambda = 1$) | 0.40 | 0.40 | **0.64** | 0.44 | 0.23 |

**Comparison with $R(x) = 0$ at Forced Terminals.** A natural alternative to flow decomposition is to simply set $R(x) = 0$ at forced terminals, correcting the target support without architectural changes. Table 5 shows that this approach degrades performance in 3 out of 4 conditions. GFN-FM $+ R(x) = 0$ collapses catastrophically on JNK3, dropping from 0.403 to 0.122, and GFN-TB $+ R(x) = 0$ also degrades on JNK3, falling from 0.624 to 0.491. While GFN-TB $+ R(x) = 0$ marginally improves on GSK3$\beta$, this gain is inconsistent and accompanied by severe degradation on the harder JNK3 task, indicating that the approach lacks reliability across objectives. This failure arises from *premature termination bias*: near-zero rewards at forced terminals propagate backward through flow conservation, suppressing flow through the entire near-boundary region and causing the model to terminate early, collapsing to short molecules. LEAKGFN avoids this by maintaining positive chemical flow $F_{\text{chem}}$ at forced terminals while using $\lambda$ to down-weight them in sampling, preserving the flow structure needed for stable training.

### 4.4. Analysis: Implicit Reachability Learning

A key claim of LEAKGFN is that $\lambda$ learns to predict reachability without explicit supervision. We provide empirical evidence in Appendix D, showing that $\lambda$ progressively learns to associate later trajectory steps with lower reachability during training, dropping from 0.93 at step 1 to 0.83 at step 8 by late training. This pattern emerges purely from flow matching without explicit reachability labels.

*Table 5.* Comparison with $R(x) = 0$ reward shaping at forced terminals. HM score (mean $\pm$ std) over 3 seeds.

| Method | JNK3 | GSK3$\beta$ |
|---|---|---|
| GFN-FM | $0.403 \pm 0.218$ | $0.716 \pm 0.071$ |
| GFN-FM $+ R(x) = 0$ | $0.122 \pm 0.093$ | $0.493 \pm 0.163$ |
| GFN-TB | $0.624 \pm 0.003$ | $0.675 \pm 0.008$ |
| GFN-TB $+ R(x) = 0$ | $0.491 \pm 0.153$ | $0.714 \pm 0.011$ |
| **LEAKGFN (ours)** | $\mathbf{0.653 \pm 0.068}$ | $\mathbf{0.760 \pm 0.008}$ |

### 4.5. Generalization to Advanced Tasks

To demonstrate the generalizability of LEAKGFN, we evaluate on two advanced molecular generation scenarios: pocket-conditioned generation and multi-objective optimization.

**Pocket-Conditioned Generation.** We integrate LEAKGFN into TacoGFN (Shen et al., 2024), a state-of-the-art pocket-conditioned molecular generation framework. Our dual-head architecture integrates as a plug-and-play module with minimal modification. Details are provided in Appendix E.

Table 6 presents results on the CrossDocked benchmark. Integrating LEAKGFN improves Vina Dock scores from $-8.24$ to $-8.61$ (mean), achieving the best binding affinity among all methods. Drug-likeness (QED), synthetic accessibility (SA), and novel hit rate also improve, demonstrating effective plug-and-play integration.

**Multi-Objective Generation.** We evaluate LEAKGFN on multi-objective molecular optimization, where the goal is to generate diverse molecules that simultaneously satisfy multiple, potentially conflicting objectives. We use the GSK3$\beta$ + JNK3 dual-inhibition task, a challenging benchmark that requires balancing activity against two distinct kinase targets. We integrate LEAKGFN into the HyperNetwork-based GFlowNet framework (HN-GFN) (Jain et al., 2023), replacing the single flow head with our dual-head architecture while preserving the preference conditioning mechanism. Performance is measured by hypervolume (HV), which quantifies the volume of objective space dominated by the Pareto front, and diversity (Div) among generated molecules. Details are provided in Appendix F.

As shown in Table 7, LEAKGFN achieves a hypervolume of 0.673, substantially outperforming the previous best method HN-GFN by 22%. This improvement indicates that our flow decomposition enables better coverage of the Pareto front, generating molecules that achieve favorable trade-offs between the two objectives. Notably, LEAKGFN maintains competitive diversity comparable to other GFlowNet methods, demonstrating that the improved Pareto coverage does not come at the cost of sample diversity.

*Table 6.* Integration with TacoGFN on pocket-conditioned generation (CrossDocked benchmark).

| | Vina Dock($\downarrow$) | | QED($\uparrow$) | | SA($\uparrow$) | | Div($\uparrow$) | | Novel hit($\uparrow$) | |
| | Mean | Median | Mean | Median | Mean | Median | Mean | Median | Mean | Median |
|---|---|---|---|---|---|---|---|---|---|---|
| Targetdiff | -7.36 | -7.56 | 0.49 | 0.49 | 0.60 | 0.59 | 0.74 | 0.73 | 0.08 | 0.05 |
| p2m | -7.49 | -7.00 | 0.57 | 0.58 | 0.75 | 0.76 | **0.74** | **0.77** | 0.07 | 0.01 |
| decompdiff | -8.35 | -8.25 | 0.37 | 0.35 | 0.56 | 0.56 | 0.6 | 0.6 | 0.45 | 0.45 |
| TacoGFN | -8.24 | -8.44 | 0.67 | 0.67 | 0.79 | 0.79 | 0.53 | 0.53 | 0.43 | 0.49 |
| Ours | **-8.61** | **-8.63** | **0.69** | **0.69** | **0.8** | **0.8** | 0.51 | 0.51 | **0.45** | **0.54** |

*Table 7.* Multi-objective generation on GSK3$\beta$ + JNK3.

| Method | HV ($\uparrow$) | Div ($\uparrow$) |
|---|---|---|
| MOEA/D* | $0.182 \pm 0.045$ | n/a |
| NSGA-III* | $0.364 \pm 0.041$ | n/a |
| PS-GFN* | $0.545 \pm 0.055$ | $0.786 \pm 0.013$ |
| Concat-GFN* | $0.534 \pm 0.069$ | $0.786 \pm 0.004$ |
| FiLM-GFN* | $0.431 \pm 0.045$ | $0.795 \pm 0.014$ |
| HN-GFN* | $0.550 \pm 0.074$ | **$0.797 \pm 0.015$** |
| Ours | **$0.673 \pm 0.044$** | $0.786 \pm 0.008$ |

*Results are taken from HN-GFN(Jain et al. (2023)).

## 5. Conclusion

We presented LEAKGFN, a dual-head architecture that addresses the fundamental problem of flow leakage in truncated GFlowNets. By decomposing flow into valid and leak components, our method learns the correct distribution over accessible molecules while explicitly accounting for probability mass that would exit the truncation boundary. The valid head implicitly learns molecular reachability through asymmetric terminal treatment, requiring no explicit supervision. Our theoretical analysis proves that this decomposition recovers the target distribution under mild assumptions. Experiments on five molecular optimization tasks demonstrate state-of-the-art performance on four tasks, with particularly strong improvements on challenging kinase inhibition targets (Table 2). The modular design enables seamless integration into existing frameworks, as demonstrated by consistent improvements when combined with TacoGFN for pocket-conditioned generation and on multi-objective optimization tasks. Several promising directions remain for future work. First, extending LEAKGFN to atom-level molecular construction would address settings where longer trajectories amplify truncation effects. Second, adapting the framework to continuous action spaces could enable applications beyond discrete molecular fragments. Finally, developing adaptive truncation strategies that dynamically adjust $L_{\max}$ during training based on the learned $\lambda$ values could further improve sample efficiency and reduce sensitivity to hyperparameter selection.

## Software and Data

Our code and trained models are publicly available at https://github.com/HwanheeKim813/LeakGFN.git. We use the fragment vocabulary and oracle models from Bengio et al. (2021), the CrossDocked benchmark (Francoeur et al., 2020) for pocket-conditioned experiments, and RDKit for molecular property computation.

## Acknowledgements

This research was supported by the National Research Foundation (NRF) funded by the Korean government (MSIT) (No. RS-2025-00523472).

## Impact Statement

This paper presents work whose goal is to advance the field of molecular generation for drug discovery. Improved molecular generation methods may accelerate the identification of drug candidates, potentially reducing the time and cost of pharmaceutical development. While our method uses computational surrogates rather than wet-lab experiments, we note that any generated molecules would require extensive safety and efficacy testing before clinical use. There are many potential societal consequences of our work, none which we feel must be specifically highlighted here.

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

# A. Implementation Details

## A.1. Model Architecture

Both the chemical head and valid head share a common state-action encoder $\theta_{\text{emb}}$, implemented using the NNCONV message passing layer from PyTorch Geometric (Zhu et al., 2024) combined with a GRU (Cho et al., 2014) for iterative refinement. The architecture processes molecular graphs with the following specifications:

**Graph Representation.** Each molecule is represented as a graph where nodes correspond to molecular fragments (building blocks) and edges represent bonds between fragments. Node features are learned embeddings indexed by fragment type, and edge features are computed as the outer product of stem-type embeddings for the connected fragments.

**Message Passing.** The encoder performs $K = 8$ message passing steps. At each step $k$:

$$m_v^{(k)} = \text{LeakyReLU}\left(\text{NNConv}(h_v^{(k-1)}, \{h_u^{(k-1)}\}_{u \in \mathcal{N}(v)}, e_{uv})\right) \tag{25}$$

$$h_v^{(k)} = \text{GRU}(m_v^{(k)}, h_v^{(k-1)}) \tag{26}$$

where $h_v^{(k)}$ is the hidden state of node $v$ at step $k$, $\mathcal{N}(v)$ denotes the neighbors of $v$, and $e_{uv}$ is the edge feature. The initial node embeddings $h_v^{(0)}$ are obtained by passing fragment embeddings through a 2-layer MLP with LeakyReLU activation.

**Dual-Head Architecture.** After message passing, the node representations are used to compute action logits:

*Chemical Head ($\theta_{chem}$):* For each stem (attachment point) $i$ at node $v$, the chemical head computes:

$$\log F_{\text{chem}}(s, a_i) = \text{MLP}_{\text{chem}}([h_v \| e_i^{\text{stem}}]) \tag{27}$$

where $e_i^{\text{stem}}$ is the stem-type embedding and $\|$ denotes concatenation. The MLP consists of 3 layers: $\text{Linear}(2d \to d) \to \text{LeakyReLU} \to \text{Linear}(d \to d) \to \text{LeakyReLU} \to \text{Linear}(d \to |\mathcal{B}|)$, where $d = 256$ is the hidden dimension and $|\mathcal{B}| = 105$ is the number of building blocks.

*Valid Head ($\theta_{valid}$):* The valid head has an identical architecture but outputs the valid fraction $\lambda$:

$$\lambda(s, a_i) = \sigma\left(\text{MLP}_{\text{valid}}([h_v \| e_i^{\text{stem}}])\right) \tag{28}$$

where $\sigma$ is the sigmoid function clamped to $[\epsilon, 1]$ with $\epsilon = 10^{-20}$ for numerical stability.

The final valid flow is computed as:

$$\log F_{\text{valid}}(s, a_i) = \log F_{\text{chem}}(s, a_i) + \log \lambda(s, a_i) \tag{29}$$

*Stop Action:* The stop action logit is computed from a global molecular representation using mean pooling:

$$\log F(s, \texttt{stop}) = \text{MLP}_{\text{stop}}\left(\frac{1}{|V|} \sum_{v \in V} h_v\right) \tag{30}$$

The stop MLP consists of 2 layers: $\text{Linear}(d \to d) \to \text{LeakyReLU} \to \text{Linear}(d \to 1)$.

## A.2. Fragment Vocabulary

We use the vocabulary of 105 molecular fragments introduced in the original GFlowNet paper (Bengio et al., 2021). The vocabulary was curated from common drug-like building blocks and includes:

- Aromatic rings: benzene, pyridine, pyrimidine, indole, purine derivatives (20 fragments)

- Aliphatic rings: cyclohexane, piperidine, piperazine, morpholine (15 fragments)

- Functional groups: amides, carboxylic acids, sulfonamides, phosphates (25 fragments)

- Linkers and small groups: methyl, ethyl, hydroxyl, halogens, cyano (45 fragments)

Each fragment has predefined attachment points (stems) where new fragments can be connected. We use the same vocabulary file (`blocks_105.json`) as Bengio et al. (2021) to ensure fair comparison with prior work.

## A.3. Self-loop Transitions

For every terminal state $s \in \mathcal{T}_{\text{valid}} \cup \mathcal{T}_{\text{forced}}$ encountered during trajectory sampling, we augment the training batch with a self-loop transition $(s, \text{stop}, s)$ labeled with reward $R(s)$. This increases the effective batch size by approximately 12.5% (one self-loop per trajectory of average length 8), but ensures accurate stop action calibration without requiring a separate auxiliary loss. The self-loop transitions are processed identically to regular transitions during the forward pass, with the terminal loss $\mathcal{L}_{\text{valid}}^{\text{term}}$ applied to enforce $F(s, \text{stop}) = R(s)$.

## A.4. Training Configuration

Table 8 summarizes the hyperparameters used across all experiments.

*Table 8.* Hyperparameters for training.

| Hyperparameter | Value |
|---|---|
| *Architecture* | |
| Hidden dimension ($d$) | 256 |
| Number of message passing steps ($K$) | 8 |
| Fragment vocabulary size ($|\mathcal{B}|$) | 105 |
| *Optimization* | |
| Optimizer | Adam |
| Learning rate | $5 \times 10^{-4}$ |
| Weight decay | 0 |
| Gradient clipping | None |
| *Training* | |
| Batch size (trajectories) | 8 |
| Total iterations | 30,000 |
| Minimum blocks ($L_{\min}$) | 2 |
| Maximum blocks ($L_{\max}$) | 8 |
| *Exploration* | |
| Random action probability ($\epsilon$) | 0.05 |
| *Loss Weights* | |
| Terminal loss coefficient ($\gamma_{\text{term}}$) | 10 |
| Chemical flow loss coefficient ($\alpha$) | 1.0 |
| Log regularization constant | Task-specific |

**Reward Transformation.** Raw oracle scores are transformed to rewards using:

$$R(x) = \left( \frac{\text{score}(x)}{\text{norm}} \right)^{\text{exp}} \tag{31}$$

where norm and exp are task-specific hyperparameters shown in Table 9.

*Table 9.* Task-specific reward transformation parameters.

| Task | norm | exp | log_reg_c |
|---|---|---|---|
| JNK3 | 0.5 | 8 | $(0.01/0.5)^8$ |
| GSK3$\beta$ | 0.2 | 12 | $(0.01/0.2)^{12}$ |
| DRD2 | 0.1 | 5 | $(0.01/0.1)^5$ |
| QED | 0.8 | 6 | $(0.01/0.8)^6$ |
| SA | 0.8 | 6 | $(0.01/0.8)^6$ |

## A.5. Oracle Models

**Kinase Inhibition (JNK3, GSK3$\beta$).**    We use Random Forest classifiers trained on bioactivity data from ExCAPE-DB (**?**). Input features are Morgan fingerprints with radius 2 and 1024 bits, computed using RDKit. The classifiers predict the probability of a molecule being active against the target kinase.

**DRD2 Activity.**    We use a Support Vector Machine (SVM) classifier trained on DRD2 activity data. Input features are Morgan fingerprints with radius 3 and 2048 bits, using count-based features.

**QED (Quantitative Estimate of Drug-likeness).**    Computed using the RDKit implementation of the QED score (Bickerton et al., 2012), which combines multiple molecular descriptors including molecular weight, logP, number of hydrogen bond donors/acceptors, polar surface area, number of rotatable bonds, and number of aromatic rings.

**SA (Synthetic Accessibility).**    Computed using the RDKit implementation of the SA score (Ertl & Schuffenhauer, 2009). We normalize the raw SA score (originally in range [1, 10]) to [0, 1] using:

$$SA_{\text{normalized}} = \frac{10 - SA_{\text{raw}}}{9} \tag{32}$$

## A.6. Evaluation Protocol

**Sampling.**    For evaluation, we sample 1,000 molecules using the learned policy without exploration ($\epsilon = 0$). We select the top-$k$ molecules ranked by reward, with $k = 100$ for all experiments.

**Metrics.**

- **Score**: Mean reward of top-$k$ molecules: Score $= \frac{1}{k}\sum_{i=1}^{k} R(x_i)$ where $x_1, \ldots, x_k$ are the top-$k$ molecules ranked by reward.

- **Diversity (Div)**:    Average pairwise Tanimoto dissimilarity among top-$k$ molecules:    Div $= 1 - \frac{2}{k(k-1)}\sum_{i<j} \text{Tanimoto}(x_i, x_j)$ where Tanimoto similarity is computed on Morgan fingerprints (radius 3, 2048 bits).

- **Uniqueness (Uni)**: Fraction of unique SMILES strings among all 1,000 sampled molecules (computed before top-$k$ selection).

- **Harmonic Mean (HM)**: Primary evaluation metric combining Score, Diversity, and Uniqueness: HM $= \frac{3}{\frac{1}{\text{Score}} + \frac{1}{\text{Div}} + \frac{1}{\text{Uni}}}$

## A.7. Computational Resources

All experiments were conducted on NVIDIA RTX 3090 GPUs (24GB). Training a single model for 30,000 iterations takes approximately 2-3 hours depending on the oracle evaluation cost. Peak GPU memory usage is approximately 4GB. We use 8 parallel sampling threads during training to improve throughput.

## A.8. Baseline Implementations

For GFlowNet baselines (GFN-FM, GFN-TB, GFN-SubTB, GFN-DB), we use the same neural network architecture and training configuration as LEAKGFN, differing only in the training objective. This ensures a fair comparison that isolates the contribution of our dual-head architecture.

- **GFN-FM**: Standard flow matching with single-head architecture.

- **GFN-TB**: Trajectory balance objective (Malkin et al., 2022) with log partition function $\log Z$ as a learnable parameter (initialized to 30, learning rate $5 \times 10^{-3}$).

- **GFN-SubTB**: Subtrajectory balance (Madan et al., 2023) with $\lambda = 0.9$ for geometric weighting of subtrajectories.

- **GFN-DB**: Dynamic batching variant that adaptively adjusts batch composition.

For non-GFlowNet baselines (Graph-GA, REINVENT, GEGL), we use results from the original papers or official implementations with default hyperparameters.

## A.9. Computational Overhead

Table 10 compares the computational cost of LEAKGFN against the GFN-FM baseline. The dual-head architecture introduces only 22.2% additional parameters, primarily from the valid head MLP, while the shared encoder $\theta_{\text{emb}}$ keeps overhead modest. Crucially, inference time increases by merely 2.8%, as both heads can be computed in parallel from the shared embeddings. This minimal overhead makes LEAKGFN practical for large-scale molecular screening campaigns where millions of candidates must be evaluated.

*Table 10.* Computational cost comparison between GFN-FM and LEAKGFN.

| Method | #Parameters | Inference Time (ms) |
|---|---|---|
| GFN-FM | 1.01M | 51.1 |
| LEAKGFN | 1.23M (+22.2%) | 52.5 (+2.8%) |

**Analysis.** The modest parameter increase stems from the architectural design where both heads share the computationally expensive graph neural network encoder. Only the final MLP layers are duplicated: the chemical head MLP and valid head MLP each contribute approximately 110K parameters. The inference overhead is even smaller because: (1) the shared encoder dominates computation time, and (2) the two head computations can be parallelized on GPU.

Compared to the performance gains achieved by LEAKGFN (e.g., +62% relative improvement on JNK3, 89% variance reduction on GSK3$\beta$), this computational overhead is negligible. The cost-effectiveness ratio strongly favors LEAKGFN for practical drug discovery applications where improved sample quality directly translates to reduced downstream experimental costs.

# B. Detailed Experimental Results

This section provides detailed breakdowns of the main experimental results, showing individual metrics (Score, Diversity, Uniqueness) that constitute the harmonic mean (HM) reported in the main text. Tables 11, 12, and 13 present the detailed metrics corresponding to Table 2.

**Analysis.** The detailed results reveal that LeakGFN's improvement in HM score primarily stems from stabilizing Uniqueness. On JNK3, GFN-FM suffers from severe Uniqueness collapse ($0.444 \pm 0.468$), indicating that the model repeatedly generates duplicate molecules—likely forced terminals. LeakGFN recovers Uniqueness to $0.930 \pm 0.063$, a $2\times$ improvement with substantially reduced variance. This validates our hypothesis that flow decomposition prevents the model from collapsing to forced terminals.

Interestingly, non-GFlowNet methods (Graph-GA, REINVENT, GEGL) achieve higher Score but suffer from low Diversity and Uniqueness, reflecting mode collapse behavior. In contrast, GFlowNet variants (DB, TB, SubTB) maintain perfect Uniqueness (1.0) but with lower Score, demonstrating successful diversity-seeking behavior at the cost of reward optimization. LeakGFN bridges this gap by maintaining high Uniqueness comparable to trajectory-level methods while achieving competitive Score through flow decomposition.

*Table 11.* Score comparison across tasks (corresponding to Table 2). All values are averaged over 3 random seeds.

| Method | JNK3 | GSK3$\beta$ | DRD2 | QED | SA |
|---|---|---|---|---|---|
| Graph-GA | $0.856 \pm 0.018$ | $0.723 \pm 0.140$ | $0.988 \pm 0.008$ | $0.931 \pm 0.004$ | $0.882 \pm 0.003$ |
| REINVENT | $0.773 \pm 0.083$ | $0.837 \pm 0.062$ | $1.00 \pm 0.000$ | $0.948 \pm 0.000$ | $0.887 \pm 0.000$ |
| GEGL | $0.685 \pm 0.272$ | $0.875 \pm 0.041$ | $1.00 \pm 0.000$ | $0.947 \pm 0.000$ | $0.892 \pm 0.001$ |
| GFN-DB | $0.397 \pm 0.002$ | $0.438 \pm 0.004$ | $0.944 \pm 0.010$ | $0.819 \pm 0.103$ | $0.726 \pm 0.012$ |
| GFN-TB | $0.624 \pm 0.003$ | $0.440 \pm 0.013$ | $0.943 \pm 0.002$ | $0.836 \pm 0.120$ | $0.734 \pm 0.007$ |
| GFN-SubTB | $0.620 \pm 0.009$ | $0.457 \pm 0.022$ | $0.717 \pm 0.089$ | $0.702 \pm 0.014$ | $0.765 \pm 0.019$ |
| GFN-FM | $0.399 \pm 0.140$ | $0.594 \pm 0.029$ | $0.661 \pm 0.146$ | $0.891 \pm 0.007$ | $0.909 \pm 0.005$ |
| **Ours** | $0.599 \pm 0.071$ | $0.626 \pm 0.043$ | $0.675 \pm 0.150$ | $0.904 \pm 0.005$ | $0.910 \pm 0.011$ |

*Table 12.* Diversity comparison across tasks (corresponding to Table 2). All values are averaged over 3 random seeds.

| Method | JNK3 | GSK3$\beta$ | DRD2 | QED | SA |
|---|---|---|---|---|---|
| Graph-GA | $0.443 \pm 0.008$ | $0.573 \pm 0.073$ | $0.717 \pm 0.062$ | $0.862 \pm 0.008$ | $0.811 \pm 0.008$ |
| REINVENT | $0.443 \pm 0.142$ | $0.484 \pm 0.049$ | $0.516 \pm 0.039$ | $0.750 \pm 0.058$ | $0.803 \pm 0.005$ |
| GEGL | $0.505 \pm 0.171$ | $0.470 \pm 0.140$ | $0.519 \pm 0.081$ | $0.835 \pm 0.019$ | $0.768 \pm 0.002$ |
| GFN-DB | $0.773 \pm 0.002$ | $0.852 \pm 0.011$ | $0.868 \pm 0.002$ | $0.898 \pm 0.004$ | $0.913 \pm 0.005$ |
| GFN-TB | $0.770 \pm 0.003$ | $0.853 \pm 0.005$ | $0.866 \pm 0.002$ | $0.887 \pm 0.007$ | $0.911 \pm 0.006$ |
| GFN-SubTB | $0.767 \pm 0.009$ | $0.846 \pm 0.011$ | $0.872 \pm 0.003$ | $0.889 \pm 0.004$ | $0.909 \pm 0.005$ |
| GFN-FM | $0.643 \pm 0.116$ | $0.807 \pm 0.021$ | $0.800 \pm 0.052$ | $0.899 \pm 0.002$ | $0.860 \pm 0.012$ |
| **Ours** | $0.541 \pm 0.061$ | $0.804 \pm 0.014$ | $0.826 \pm 0.037$ | $0.884 \pm 0.004$ | $0.888 \pm 0.010$ |

*Table 13.* Uniqueness comparison across tasks (corresponding to Table 2). All values are averaged over 3 random seeds.

| Method | JNK3 | GSK3$\beta$ | DRD2 | QED | SA |
|---|---|---|---|---|---|
| Graph-GA | $0.430 \pm 0.020$ | $0.458 \pm 0.031$ | $0.479 \pm 0.023$ | $0.501 \pm 0.016$ | $0.411 \pm 0.009$ |
| REINVENT | $0.608 \pm 0.054$ | $0.622 \pm 0.087$ | $0.378 \pm 0.077$ | $0.525 \pm 0.137$ | $0.315 \pm 0.006$ |
| GEGL | $0.793 \pm 0.038$ | $0.670 \pm 0.171$ | $0.777 \pm 0.069$ | $0.834 \pm 0.018$ | $0.493 \pm 0.015$ |
| GFN-DB | $1.000 \pm 0.000$ | $1.000 \pm 0.000$ | $1.000 \pm 0.000$ | $1.000 \pm 0.000$ | $1.000 \pm 0.000$ |
| GFN-TB | $1.000 \pm 0.000$ | $1.000 \pm 0.000$ | $1.000 \pm 0.000$ | $1.000 \pm 0.001$ | $1.000 \pm 0.001$ |
| GFN-SubTB | $1.000 \pm 0.001$ | $1.000 \pm 0.001$ | $1.000 \pm 0.000$ | $1.000 \pm 0.000$ | $0.909 \pm 0.005$ |
| GFN-FM | $0.444 \pm 0.468$ | $0.814 \pm 0.206$ | $0.939 \pm 0.027$ | $0.996 \pm 0.001$ | $0.970 \pm 0.009$ |
| **Ours** | $0.930 \pm 0.063$ | $0.910 \pm 0.044$ | $0.962 \pm 0.021$ | $1.000 \pm 0.000$ | $0.975 \pm 0.017$ |

## C. Sensitivity Analysis on $\alpha$

We investigate the interaction between the loss weight $\alpha$ and the maximum trajectory length $L_{\max}$ on DRD2. This analysis tests our hypothesis that the chemical head becomes more important as truncation effects intensify.

*Table 14.* Interaction between $\alpha$ and $L_{\max}$ on DRD2. HM score (mean) over 3 random seeds. Higher is better.

| | $L_{\max} = 6$ | $L_{\max} = 8$ | $L_{\max} = 10$ | $L_{\max} = 12$ |
|---|---|---|---|---|
| GFN-FM | $0.863 \pm 0.038$ | $0.775 \pm 0.041$ | $0.356 \pm 0.151$ | $0.380 \pm 0.241$ |
| $\alpha = 0$ | $0.825 \pm 0.044$ | $0.801 \pm 0.035$ | $0.591 \pm 0.086$ | $0.350 \pm 0.037$ |
| $\alpha = 0.1$ | $0.877 \pm 0.025$ | $0.829 \pm 0.122$ | $0.574 \pm 0.115$ | $0.528 \pm 0.068$ |
| $\alpha = 0.5$ | $0.880 \pm 0.013$ | $0.780 \pm 0.054$ | $0.539 \pm 0.103$ | $0.355 \pm 0.052$ |
| $\alpha = 1.0$ | $0.865 \pm 0.022$ | $0.797 \pm 0.070$ | $0.517 \pm 0.086$ | $0.435 \pm 0.078$ |
| $\alpha = 2.0$ | $0.895 \pm 0.013$ | $0.260 \pm 0.402$ | $0.344 \pm 0.127$ | $0.466 \pm 0.072$ |

**Effect of $\alpha$.** When $\alpha = 0$, the chemical head receives no direct supervision and is trained only through gradients backpropagated from $\mathcal{L}_{\text{valid}}$. Compared to GFN-FM, even $\alpha = 0$ provides consistent improvements at larger $L_{\max}$ (e.g., 0.591 vs 0.356 at $L_{\max} = 10$), demonstrating that the dual-head architecture itself contributes to stability regardless of explicit chemical flow supervision.

**However, excessively large $\alpha$ can destabilize training.** At $\alpha = 2.0$ with $L_{\max} = 8$, we observe a significant performance drop ($0.260 \pm 0.402$). This instability arises from *conflicting gradient signals* between the two loss terms: $\mathcal{L}_{\text{valid}}$ encourages $F_{\text{valid}}(s, a) \to 0$ at forced terminals, while $\mathcal{L}_{\text{chem}}$ maintains $F_{\text{chem}}(s, a) > 0$. When $\alpha$ is large, the chemical loss dominates, disrupting the delicate balance required for $\lambda$ to learn correct reachability estimates.

**Interaction with $L_{\max}$.** At small $L_{\max}$ (e.g., 6), all configurations perform similarly well, as truncation effects are minimal. As $L_{\max}$ increases, the gap between LeakGFN variants and GFN-FM widens substantially. Notably, $\alpha = 2.0$

exhibits instability at $L_{\max} = 8$ ($0.260 \pm 0.402$), suggesting that overweighting the chemical loss can destabilize training when valid and chemical flows diverge significantly. Moderate values ($\alpha \in [0.1, 1.0]$) provide the best balance between leveraging chemical flow information and maintaining stable optimization.

**Gradient Flow Analysis.** A potential concern with the multiplicative decomposition $F_{\text{valid}} = F_{\text{chem}} \cdot \lambda$ is gradient starvation: when $\lambda \to 0$ near the truncation boundary, gradients from $\mathcal{L}_{\text{valid}}$ to $F_{\text{chem}}$ diminish since $\partial F_{\text{valid}} / \partial F_{\text{chem}} = \lambda$. We address this through two mechanisms:

*Direct supervision via $\mathcal{L}_{chem}$.* The chemical head loss provides gradients to $F_{\text{chem}}$ independent of $\lambda$, ensuring continued learning even when $\lambda \approx 0$. This motivates using $\alpha > 0$.

*Shared encoder gradients.* Even when $\lambda \to 0$, gradients flow to the shared encoder $\theta_{\text{emb}}$ through the valid head's $\lambda$ branch: $\partial \mathcal{L}_{\text{valid}} / \partial \theta_{\text{emb}}$ includes terms proportional to $F_{\text{chem}} \cdot (\partial \lambda / \partial \theta_{\text{emb}})$, which remain non-zero.

Empirically, we observe stable training across $\alpha \in [0, 1]$ in Table 14. The instability at $\alpha = 2.0$ arises from *conflicting gradient signals*—$\mathcal{L}_{\text{valid}}$ pushing $F_{\text{valid}}(s, a) \to 0$ at forced terminals while $\mathcal{L}_{\text{chem}}$ maintains $F_{\text{chem}}(s, a) > 0$—rather than from gradient starvation.

**Key Finding: Architecture vs. Supervision.** A surprising finding is that $\alpha = 0$ (no explicit chemical supervision) performs competitively at larger $L_{\max}$ values. This suggests that the *architectural* contribution of the dual-head decomposition— enabling implicit reachability learning through $\lambda$—is more fundamental than the explicit chemical flow supervision. When $L_{\max}$ is large, the chemical space becomes vast and the chemical head must model flow over regions that are never visited during training. In this regime, explicit supervision via $\mathcal{L}_{\text{chem}}$ may introduce noise, while implicit learning through backpropagation from $\mathcal{L}_{\text{valid}}$ adapts more flexibly.

This finding has important implications for understanding LEAKGFN's mechanism: the core contribution is not the chemical flow supervision itself, but rather the *multiplicative decomposition* $F_{\text{valid}} = F_{\text{chem}} \cdot \lambda$ that enables $\lambda$ to learn reachability implicitly. The chemical head serves primarily as a "base flow" that $\lambda$ modulates, and this modulation can be learned effectively through gradients from the valid loss alone.

**Variance Reduction.** Across all settings, LeakGFN variants exhibit substantially lower variance than GFN-FM, particularly at challenging $L_{\max}$ values (e.g., std 0.086 vs 0.151 at $L_{\max} = 10$). Even $\alpha = 0$ achieves this variance reduction, confirming that the dual-head architecture provides implicit regularization independent of the chemical loss weight.

**Practical Recommendation.** Based on this analysis, we provide the following guidelines for practitioners:

- Use $\alpha \in [0.1, 0.5]$ when $L_{\max}$ is set close to the expected molecule size distribution.

- Use $\alpha = 0$ when $L_{\max}$ is set conservatively large or when the target molecule size is unknown.

- Avoid $\alpha \geq 2.0$ due to potential training instability, especially at moderate $L_{\max}$ values.

- When in doubt, $\alpha = 1$ provides a reasonable default with consistent performance across most settings.

## D. Implicit Reachability Learning Analysis

Figure 2 provides direct evidence that $\lambda$ learns to predict reachability without explicit supervision by visualizing the learned valid fraction across trajectory steps during training.

We fix a set of length-8 trajectories and measure $\lambda$ at each step across training checkpoints. Curves show the mean $\lambda$ averaged over checkpoints within each training stage. Early in training (0-10K), $\lambda$ incorrectly increases with trajectory step due to random initialization. During mid training (10K-20K), $\lambda$ becomes flat as the model transitions. By late training (20K-30K), $\lambda$ exhibits the correct decreasing trend, dropping from 0.93 at step 1 to 0.83 at step 8. This pattern emerges purely from flow matching without explicit reachability labels, confirming that the asymmetric terminal treatment successfully induces implicit reachability learning.

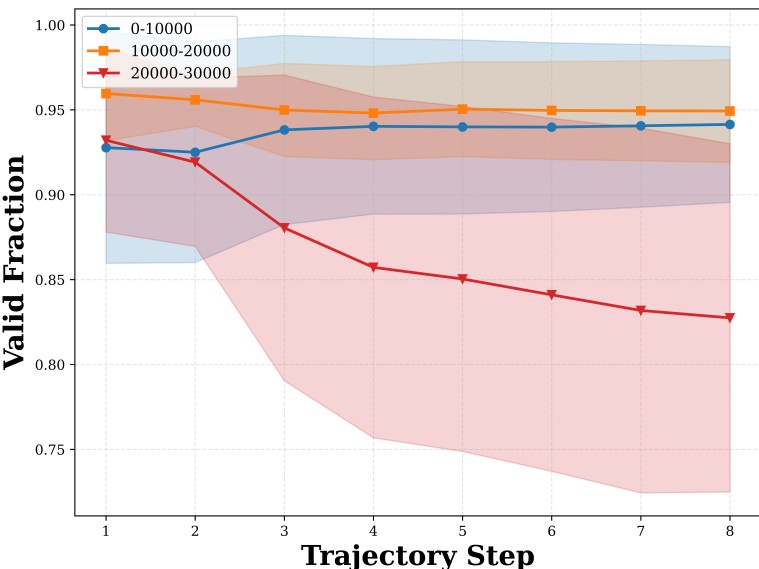

*Figure 2.* Evolution of learned valid fraction $\lambda$ during training on JNK3 ($L_{\max} = 8$). The model progressively learns to associate later trajectory steps with lower reachability.

## E. Pocket-Conditioned Generation Setup

This section provides detailed experimental setup for the pocket-conditioned generation experiments in Section 4.5.

**Setup.** TacoGFN generates molecules conditioned on protein binding pockets by combining a pocket encoder with fragment-based GFlowNet generation. Like other fragment-based approaches, TacoGFN requires trajectory truncation, making it susceptible to flow leakage. Our dual-head architecture integrates as a plug-and-play module: we replace TacoGFN's single flow prediction head with our chemical and valid heads while keeping the pocket encoder and fragment embeddings unchanged. This requires minimal architectural modification.

**Baselines.** We compare against state-of-the-art structure-based drug design methods:

- **Diffusion-based approaches:** TargetDiff (Guan et al., 2023) and DecompDiff (Guan et al., 2024)

- **Autoregressive methods:** Pocket2Mol (P2M) (Peng et al., 2022)

- **GFlowNet baseline:** The original TacoGFN without flow decomposition

**Metrics.** We evaluate on the CrossDocked benchmark (Francoeur et al., 2020) using five metrics:

- **Vina Dock:** Predicted binding affinity (kcal/mol) using AutoDock Vina. Lower values indicate stronger predicted binding.

- **QED:** Quantitative estimate of drug-likeness (Bickerton et al., 2012), ranging from 0 to 1.

- **SA:** Synthetic accessibility score, normalized to $[0, 1]$ where higher is more synthetically accessible.

- **Diversity:** Average pairwise Tanimoto dissimilarity (on Morgan fingerprints, radius 2, 2048 bits) among generated molecules for each pocket.

- **Novel hit rate:** Fraction of generated molecules satisfying: (i) Vina Dock $< -8.0$ kcal/mol, and (ii) maximum Tanimoto similarity to any training molecule $< 0.4$. This metric measures the ability to discover novel, high-affinity binders.

# F. Multi-Objective Generation Setup

This section provides detailed experimental setup for the multi-objective generation experiments in Section 4.5.

**Task.** We evaluate on the GSK3$\beta$ + JNK3 dual-objective task, where the goal is to generate molecules that simultaneously inhibit both kinases. This is a challenging multi-objective optimization problem as the two objectives may have conflicting structural requirements.

**Integration with HN-GFN.** We adopt the HyperNetwork-based preference conditioning framework of HN-GFN (Jain et al., 2023) and integrate our dual-head architecture as a drop-in replacement for the flow prediction module. Specifically:

- **Preference conditioning:** A preference vector $\omega \in \Delta^{K-1}$ (where $K = 2$ for dual objectives) is sampled uniformly during training. The hypernetwork generates weights for both the chemical head and valid head conditioned on $\omega$.

- **Reward scalarization:** Following HN-GFN, we use the weighted sum $R(x; \omega) = \sum_{k=1}^{K} \omega_k R_k(x)$ where $R_k(x)$ is the reward for objective $k$.

- **Flow decomposition:** Our dual-head decomposition $F_{\text{valid}} = F_{\text{chem}} \cdot \lambda$ is applied independently for each preference, enabling the model to learn reachability patterns that may vary across different regions of the Pareto front.

All other hyperparameters (learning rate, batch size, network architecture) follow the original HN-GFN implementation to ensure fair comparison.

**Baselines.** We compare against both evolutionary and GFlowNet-based multi-objective methods:

- **Evolutionary algorithms:** MOEA/D and NSGA-III, classical multi-objective optimization methods

- **GFlowNet variants:**
    - PS-GFN: Preference-conditioned GFlowNet trained separately per preference (gold standard)
    - Concat-GFN: Concatenates preference vector to state representation
    - FiLM-GFN: Uses Feature-wise Linear Modulation for preference conditioning
    - HN-GFN: HyperNetwork-based preference conditioning (Jain et al., 2023)

Results for baselines are taken directly from Jain et al. (2023) to ensure consistent evaluation protocols.

**Metrics.** We evaluate multi-objective performance using the following metrics:

- **Hypervolume (HV):** Volume of the objective space dominated by the Pareto front, computed with reference point $(0, 0)$. Higher values indicate better coverage and quality of the Pareto front. Formally, $\text{HV} = \text{Vol}(\{y \in \mathbb{R}^K : \exists x \in \mathcal{P}, y \preceq f(x)\})$ where $\mathcal{P}$ is the Pareto front and $f(x) = (R_1(x), R_2(x))$.

- **Diversity (Div):** Average pairwise Tanimoto dissimilarity (on Morgan fingerprints, radius 3, 2048 bits) among molecules on the Pareto front. This measures structural diversity of the discovered trade-off solutions.

# G. Generated Molecules

We present representative molecules generated by LEAKGFN on each optimization task. For each task, we show the top-scoring molecules along with their oracle scores and key molecular properties.

### G.1. JNK3 Kinase Inhibition

Figure 3 shows the top-6 molecules generated by LEAKGFN for JNK3 kinase inhibition. These molecules exhibit structural features commonly associated with kinase inhibitors, including aromatic rings and hydrogen bond acceptors.

### G.2. GSK3$\beta$ Kinase Inhibition

Figure 4 shows the top-6 molecules generated for GSK3$\beta$ inhibition.

### G.3. DRD2 Activity

Figure 5 shows the top-6 molecules generated for DRD2 receptor activity.

### G.4. Drug-likeness (QED)

Figure 6 shows molecules optimized for drug-likeness (QED score).

### G.5. Synthetic Accessibility (SA)

Figure 7 shows molecules optimized for synthetic accessibility (SA score).

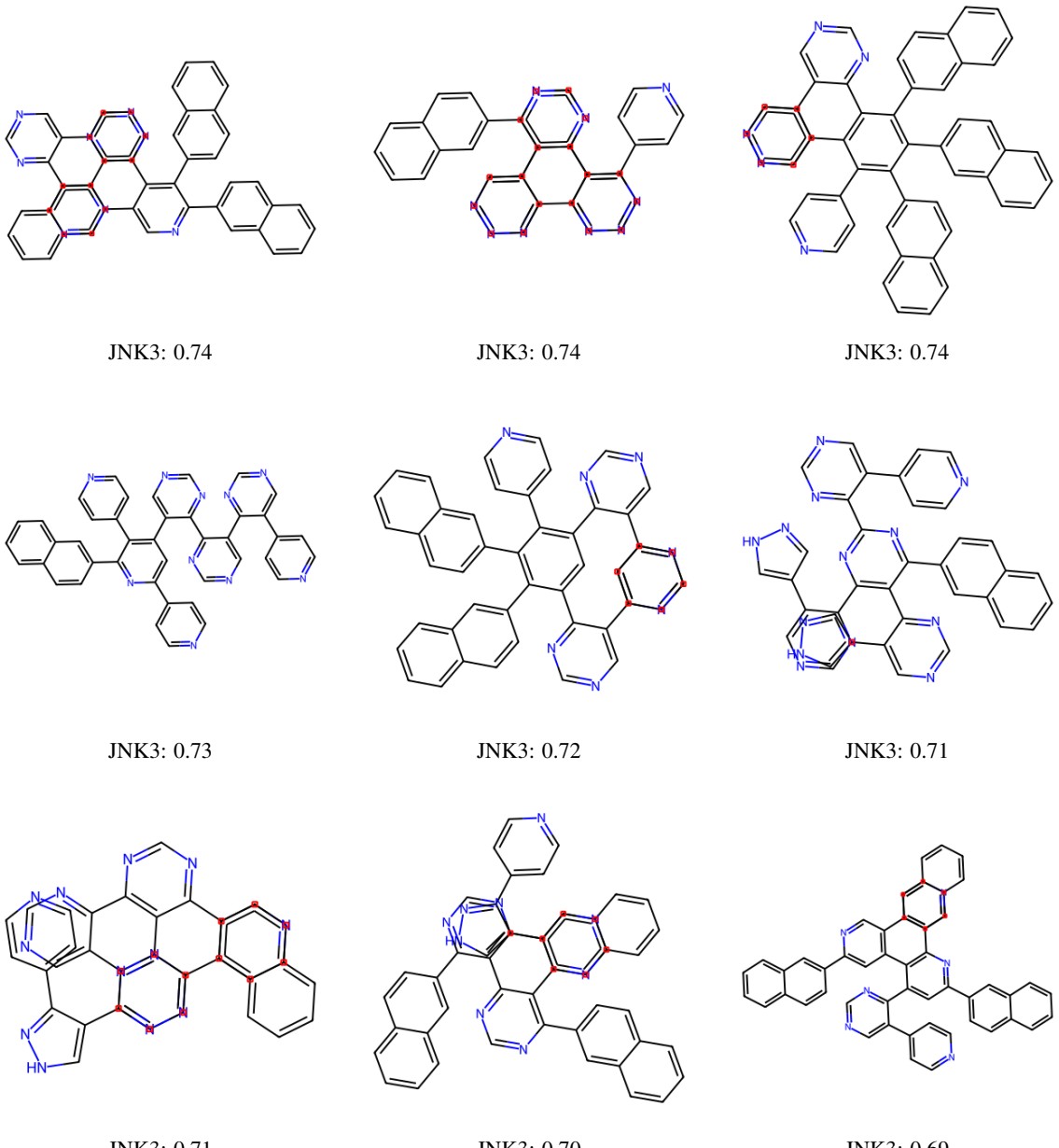

*Figure 3.* Top-6 molecules generated by LEAKGFN for JNK3 kinase inhibition. Each molecule is shown with its predicted activity score and QED value.

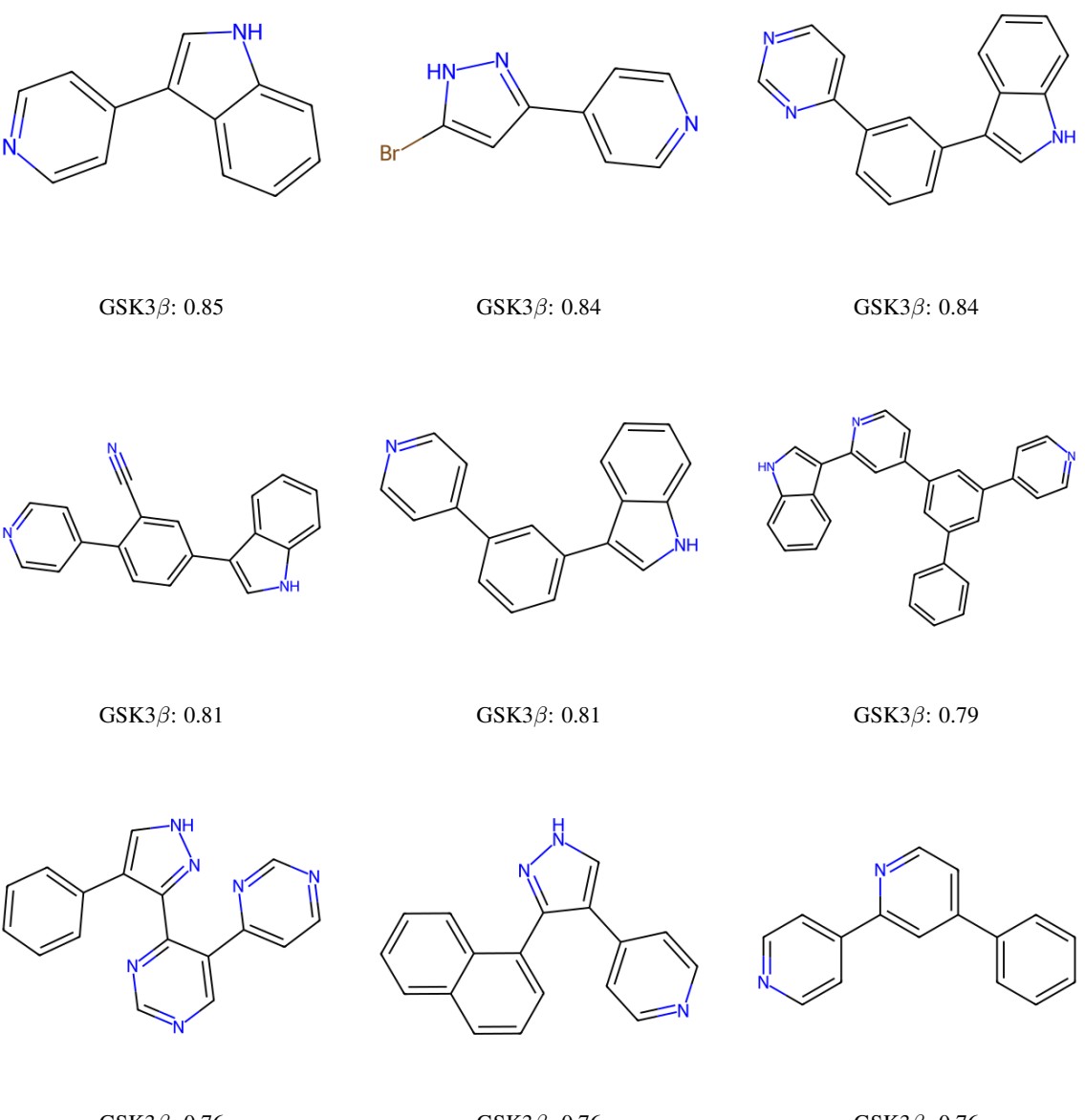

*Figure 4.* Top-6 molecules generated by LEAKGFN for GSK3$\beta$ kinase inhibition.

DRD2: 0.895          DRD2: 0.885          DRD2: 0.861

DRD2: 0.856          DRD2: 0.838          DRD2: 0.833

DRD2: 0.803          DRD2: 0.771          DRD2: 0.771

*Figure 5.* Top-6 molecules generated by LEAKGFN for DRD2 receptor activity.

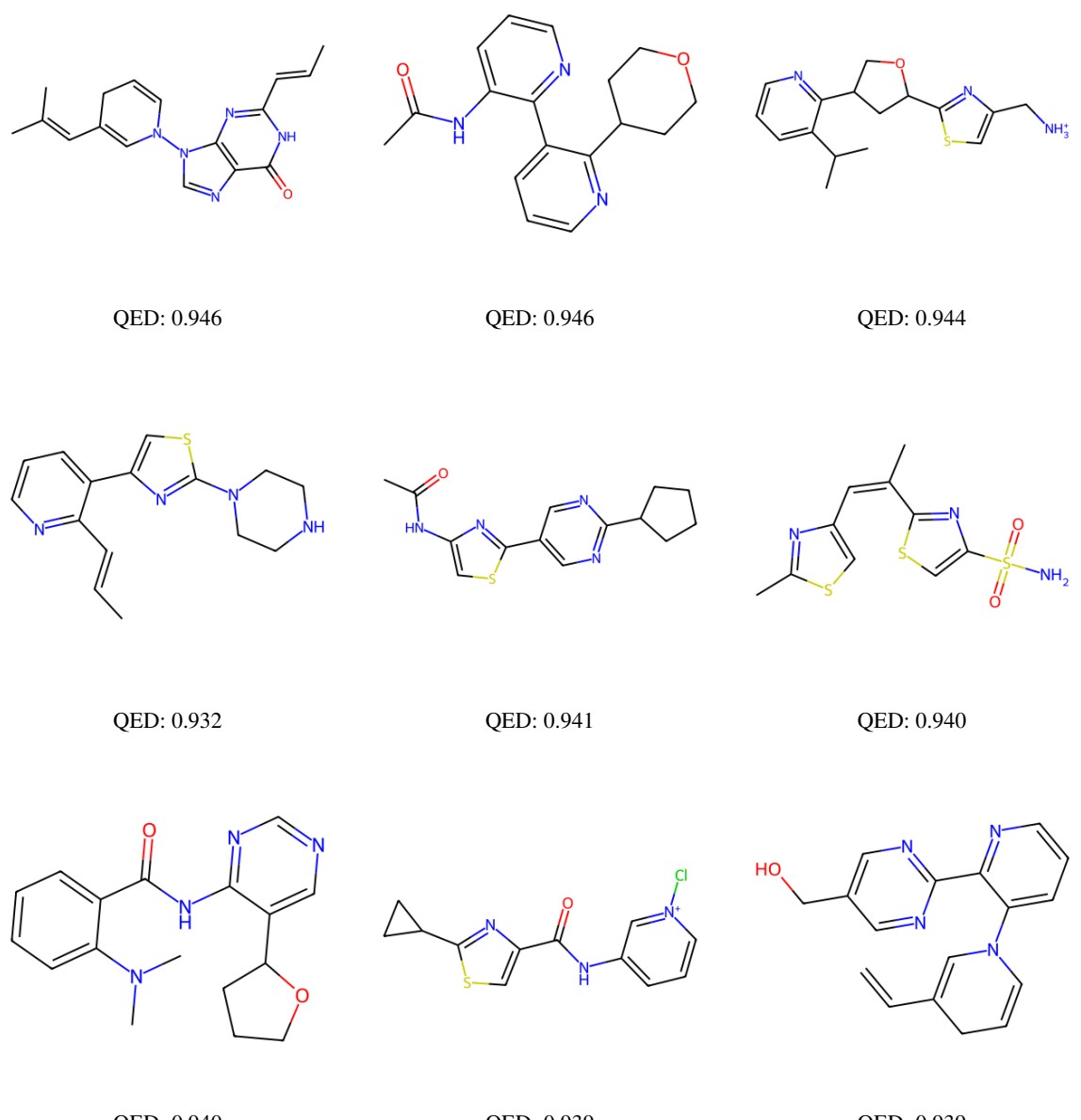

QED: 0.946                    QED: 0.946                    QED: 0.944

QED: 0.932                    QED: 0.941                    QED: 0.940

QED: 0.940                    QED: 0.939                    QED: 0.939

*Figure 6.* Top-6 molecules generated by LEAKGFN for drug-likeness optimization.

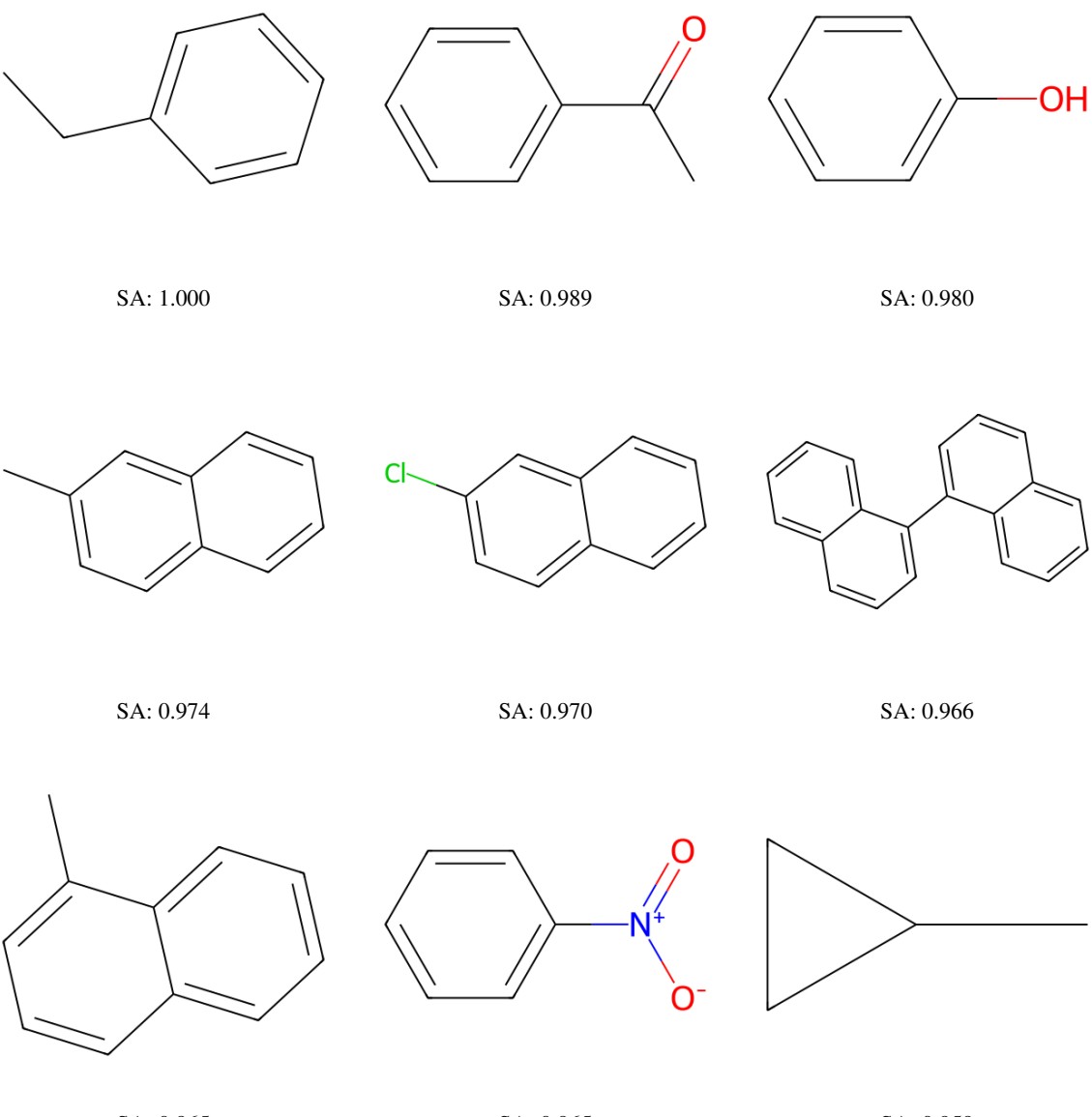

SA: 1.000                SA: 0.989                SA: 0.980

SA: 0.974                SA: 0.970                SA: 0.966

SA: 0.965                SA: 0.965                SA: 0.959

*Figure 7.* Top-6 molecules generated by LEAKGFN for drug-likeness optimization.

