# OpenReview forum: "LeakGFN: Robust Molecular Generation in Generative Flow Networks via Flow Decomposition"
_ICML.cc/2026/Conference — ICML 2026 regular_

### Official Review · Reviewer_4DnB · 2026-02-19

**Soundness:** 2
**Presentation:** 2
**Significance:** 2
**Originality:** 3
**Overall Recommendation:** 3
**Confidence:** 3

**Summary:**

The paper introduces an architecture that decomposes flow into two components: “valid” that estimates the flow reaching molecules which generation terminated before the maximum number of steps, and “chemical” that estimates the flow for the entire, non-truncated chemical space. Moreover, authors propose an exploration strategy that instead of taking actions at uniform, follows the “chemical” flow. The new flow parametrization along with the new exploration strategy improves the average reward of the sampled molecules and the harmonic mean of reward, uniqueness, and diversity, while reducing the diversity itself.

**Compliance With Llm Reviewing Policy:**

Affirmed.

**Final Justification:**

Since my previous questions remain unaddressed (likely because of tight deadlines), I am still seeking clarity on an important assumption of the paper: is $T_valid \cup T_{forced}  = \emptyset$? While this seems to follow implicitly from the text, the underperformance of R(x)=0 leaves me somewhat uncertain. The manuscript would certainly benefit from a more precise formulation of the problem.

I am raising my score to 3 while lowering my confidence to account for the remaining ambiguity. My final recommendation remains neutral. Ideally, my numerical score should not be the primary factor in the final decision; rather, the conceptual concerns I have raised deserve careful consideration.

**Key Questions For Authors:**

See weaknesses.

**Limitations:**

There's no limitation section.

**Strengths And Weaknesses:**

### Strengths:
1. The paper is easy to follow.
2. The evaluation is done on multiple datasets.
3. The method is a plug-in and can be used with existing GFlowNets.

### Weaknesses:
1. Authors claim that truncation fundamentally violates the flow conservation principle, which is not justified. The truncation is one of the components defining the DAG underlying GFlowNet: it constitutes the reachable molecule space. And within that space, properly trained GFlowNet respects the flow conservation. Using a fragment-based approach also limits the reachable molecule space, but similarly to truncation, it does not prohibit the flow conservation.
2. The definition of a “validity” of the state requires a deep elaboration. Why are the forced terminals “invalid” and other states are “valid”? Why are we interested in such a notion of “validity”?
3. It appears that assigning R(x)=0 to invalid states would achieve the same result as the proposed method (as suggested in Section 3.6). Could the authors explain the specific advantages of the dual-head architecture over this simpler reward-shaping approach?
4. Authors do not decouple the influence of their new exploration policy and the dual-head architecture. Intuitively, their exploration policy is less “exploratory” than the standard $\epsilon$-exploration, which may largely contribute to higher reward and lower diversity of their approach. Those components must be decoupled for meaningful evaluation.

The paper has major flaws in both problem formulation and experimental design. The changes required to fix the paper are fundamental and cannot be done in the rebuttal period. Therefore, I opt for rejecting the paper.

---

> ### Author Rebuttal · Authors · 2026-03-31
>
> We thank the reviewer for the detailed feedback. We address each concern below and hope to clarify our problem formulation.
>
> **Q1. Truncation does not violate flow conservation.**
>
> We agree that flow conservation holds within the truncated DAG. We will revise our language to avoid this confusion. Our claim is about target distribution mismatch, not flow conservation violation.
>
> The key distinction is between truncation and fragment vocabulary. The fragment vocabulary limits which molecules can be constructed*, but every terminal under vocabulary restriction is a voluntary stop: the agent chose to terminate, and the resulting molecule is its intentional output. Truncation is qualitatively different: it limits *how far along a construction trajectory the model can go*, and creates terminals where the agent did not choose to stop. The resulting forced terminals are valid molecular graphs (as we note in Section 3.4), but the agent would have continued if allowed. Flow conservation over this truncated DAG distorts the learned policy: the model over-concentrates on paths that fit within $L_{max}$, losing diversity.
>
> Standard GFlowNets learn $P(x) \propto R(x)$ over $\mathcal{T}\_{valid} \cup \mathcal{T}\_{forced}$. The intended target is $P(x) \propto R(x)$ over $\mathcal{T}\_{valid}$ only. This mismatch is empirically significant: on JNK3, GFN-FM's Uniqueness collapses to 0.444±0.468 (Appendix B), while LeakGFN recovers to 0.930±0.063. The [Uniqueness over training iterations](https://anonymous.4open.science/r/LeakGFN-464A/figures/uniqueness_over_iterations.png) shows this collapse occurring dynamically during training.
>
> **Q2. Why are forced terminals "invalid"?**
>
> The distinction is **voluntary vs. forced termination**. GFlowNet's flow conservation assumes termination is the agent's decision — the agent chooses the stop action when it considers the molecule complete. Forced terminals violate this premise: termination is imposed by the environment ($L_{max}$), not chosen by the agent. The agent may have preferred to continue construction.
>
> This matters because the resulting distribution includes states where generation was artificially halted, diluting probability mass that should be allocated exclusively to intentional outputs. Even with small individual rewards at forced terminals (typically R(s)<0.1), their aggregate effect distorts relative probabilities among valid molecules — as demonstrated by the Uniqueness collapse on JNK3/GSK3$\beta$.
>
> **Q3. Why not simply set R(x)=0 at forced terminals?**
>
> Two reasons. First, **gradient vanishing**: setting R(x)=0 pushes $F_{in}(x) \to 0$, after which these trajectories contribute near-zero gradients to the shared encoder (analogous to dying ReLU). Excluding them from the loss entirely wastes training samples. This creates a dilemma — oracle reward R(s)>0 preserves gradients but pollutes the target support; R(x)=0 corrects the support but starves the model.
>
> Second, and more fundamentally, R(x)=0 only tells the model to *avoid* forced terminals. It provides no mechanism to prevent flow distortion near the truncation boundary. When truncation cuts outgoing paths from a state, flow conservation forces the remaining paths to absorb all the flow, causing over-concentration and mode collapse. LeakGFN's $F_{chem}$ addresses this by maintaining positive outgoing flow at forced terminals (treating them as non-terminals), preventing this distortion near the boundary and preserving policy diversity.
>
> Together: $F_{chem}$ preserves gradients *and* prevents flow distortion near the boundary, while $\lambda$ down-weights forced terminals in sampling. Our $\alpha=0$ analysis (Appendix C) confirms: even without explicit chemical flow supervision, the decomposition alone improves over GFN-FM (0.591 vs 0.356 at $L_{max}=10$ on DRD2).
>
> **Q4. Exploration and dual-head are not decoupled.**
>
> Table 4 provides a comparison that isolates each contribution:
>
> | Setting | HM (JNK3) |
> |---|---|
> | Single-head + $\epsilon$-exploration (GFN-FM) | 0.403 |
> | Dual-head + $\epsilon$-exploration (w/o $\pi_{explore}$) | 0.62 |
> | Dual-head + flow-guided (LeakGFN full) | 0.64 |
>
> The dual-head architecture alone accounts for the dominant improvement (+0.22 over GFN-FM), while flow-guided exploration adds a modest further gain (+0.02). This confirms that our primary contribution — the flow decomposition — is responsible for the performance improvement, independent of the exploration strategy.

---

> > ### Author Rebuttal · Reviewer_4DnB · 2026-04-03
> >
> > Q1. If the intention is to learn $P(x)$ over only valid states, we can set $R(x)=0$ on forced terminals. This would be mathematically equivalent to the LeakGFN formulation. Comparing against such a basic baseline is a necessity for the paper to be accepted. Currently, other baselines learn different distribution than LeakGFN, which results in an "apples-to-oranges" comparison.
> >
> > Q2. From this response and other rebuttals, I see that a potential advantage of LeakGFN is that the underlying policy could converge faster because it is trained as if all valid molecules are accessible. However, again, this requires a comparison to baselines that learn the same distribution (i.e., setting $R(x)=0$ on forced terminals).
> >
> > Q3. Standard GFlowNet objectives are computed in log-space, so the argument regarding vanishing gradients does not apply here. The second argument is more convincing, and I believe it can be framed as a potential advantage of LeakGFN, as mentioned above.
> >
> > Q4. Table 4 indeed shows the decoupled contributions of leak flow and novel exploration. I missed it; mea culpa.
> >
> > To summarize: a comparison to all other training objectives with $R(x)=0$ is a strict necessity for the paper to be accepted. Only then we will be able to observe a true advantage of LeakGFN over a trivial change of the sampling distribution from $P(x)$ over $T_{valid} \cup T_{forced}$ to $P(x)$ over $T_{valid}$.

---

> > > ### Author Response · Authors · 2026-04-05
> > >
> > > We sincerely thank Reviewer 4DnB for the constructive follow-up.
> > >
> > > **Q1 & Q2. Comparison with $R(x)=0$ baseline**
> > >
> > > As suggested, we set $R(x)=0$ for forced terminals (clipped to a small constant for numerical stability, all other components identical) and evaluated on JNK3 and GSK3$\beta$ with both FM and TB.
> > >
> > > | Method | JNK3 (HM) | GSK3$\beta$ (HM) |
> > > |---|---|---|
> > > | GFN-FM | 0.403 ± 0.218 | 0.716 ± 0.071 |
> > > | GFN-FM + $R(x)=0$ | 0.122 ± 0.093 | 0.493 ± 0.163 |
> > > | GFN-TB | 0.624 ± 0.003 | 0.675 ± 0.008 |
> > > | GFN-TB + $R(x)=0$ | 0.491 ± 0.153 | 0.714 ± 0.011 |
> > > | LeakGFN (ours) | **0.653 ± 0.068** | **0.760 ± 0.008** |
> > >
> > > $R(x)=0$ degrades performance in 3 out of 4 conditions. FM + $R(x)=0$ suffers catastrophic collapse on JNK3 (0.403 → 0.122) and substantial degradation on GSK3$\beta$ (0.716 → 0.493). TB + $R(x)=0$ also degrades on JNK3 (0.624 → 0.491). The only marginal improvement is TB on GSK3$\beta$ (0.675 → 0.714), which remains far below LeakGFN (0.760). $R(x)=0$ also exhibits high variance (e.g., ±0.153 for TB JNK3), compared to LeakGFN's stable performance (±0.068 and ±0.008).
> > >
> > > We identify two reasons for this failure:
> > >
> > > (1) **Premature termination bias.** When forced terminals receive near-zero reward, the model avoids reaching $L_{\max}$ by terminating early, collapsing generation to a small set of short molecules. This replaces the original $L_{\max}$ bias with the opposite pathology. This effect is especially severe for FM, where the near-zero reward signal propagates backward through state-level flow conservation, suppressing flow through the entire near-boundary region.
> > >
> > > (2) **Inconsistent behavior across objectives and tasks.** $R(x)=0$ causes catastrophic failure for FM but only moderate degradation for TB, and improves TB on GSK3$\beta$ while degrading it on JNK3. This inconsistency demonstrates that $R(x)=0$ is not a principled solution — its effect depends unpredictably on the interaction between the training objective and the reward landscape.
> > >
> > > LeakGFN avoids both failure modes. $F_{\text{chem}}$ preserves the flow structure over the full chemical space, while $\lambda$ learns reachability gradually without distorting the reward signal. The divergent behavior across objectives also confirms that these results reflect genuine optimization dynamics rather than implementation artifacts.
> > >
> > > **Q3. Log-space and vanishing gradients**
> > >
> > > We agree with the reviewer's clarification that standard GFlowNet objectives operate in log-space. As the reviewer noted, the support pollution argument is the more fundamental concern. Our experiments above reinforce this: $R(x)=0$ fails because it eliminates useful learning signal near the truncation boundary, preventing the model from discovering valid molecules that require longer trajectories.
> > >
> > > These experiments confirm that while the reviewer's insight about forced terminal treatment is correct, $R(x)=0$ is not a reliable solution (degrading 3 out of 4 conditions), whereas LeakGFN's flow decomposition consistently achieves the best performance. We will extend to remaining tasks in the camera-ready version and welcome any further feedback.

---

### Official Review · Reviewer_Zutc · 2026-03-05

**Soundness:** 3
**Presentation:** 2
**Significance:** 3
**Originality:** 2
**Overall Recommendation:** 4
**Confidence:** 4

**Summary:**

This paper proposes LeakGFN, a modification to Generative Flow Networks (GFlowNets) for molecular optimization under trajectory truncation. The key idea is to decompose the flow into two components using a dual-head architecture: a chemical head that models the full chemical flow and a valid head that estimates the fraction of flow that reaches valid molecules within the truncation boundary. The authors argue that standard GFlowNet training allocates probability mass to incomplete molecular fragments when trajectories are truncated, which they describe as “flow leakage”. The proposed method aims to correct this issue by separating valid flow from leaked flow during training and sampling. Experiments on several molecular optimization benchmarks show improvements over flow-matching based GFlowNet baselines.

**Compliance With Llm Reviewing Policy:**

Affirmed.

**Final Justification:**

The authors solved my concerns and promised to update the revision according to the rebuttal. I believe it will make the manuscript stronger for publication. So I raise my score.

**Key Questions For Authors:**

1. Lack of convincing evidence for the core phenomenon.

The central motivation relies on the assumption that trajectory truncation leads to a significant number of invalid or incomplete molecules being treated as terminal states. However, the paper does not provide sufficient empirical evidence demonstrating that this phenomenon widely occurs in practical molecular optimization settings. In many implementations, the maximum trajectory length is chosen sufficiently large to ensure full molecule construction, which may substantially reduce the frequency of truncated fragments. Without quantitative analysis, it is difficult to assess the practical severity of the proposed “flow leakage” issue.

2. Unclear definition of “flow leakage”.

The term “flow leakage” is used throughout the paper as a central concept, but its definition remains vague. It is not clearly formalized whether this refers to probability mass assigned to forced terminal states, flow leaving the accessible molecular space, or a mismatch between the learned and target distributions. Additionally, the paper does not provide clear examples of practical failure cases caused by this phenomenon. A more precise definition and concrete empirical demonstrations would improve clarity.

3. Triviality of Theorem 3.1.

Theorem 3.1 appears to be trivial and does not demonstrate a novel theoretical property of the proposed method. The result essentially states that if the learned flow satisfies the standard flow-matching conditions, then the model recovers the target reward-proportional distribution. However, this property is already guaranteed by the standard GFlowNet framework under ideal convergence. And this work also can not provide formal convergence guarantees. Therefore, the theorem does not help establish the novelty of the method, and it's property can be directly obtained via the usage of GFlowNet framework.

4. Limited comparison with recent methods.

The experimental section compares mainly with earlier GFlowNet variants and several classical baselines. However, the most recent baseline methods appear to be from around 2023, while molecular optimization have progressed rapidly in recent years. The absence of comparisons with more recent approaches makes it difficult to judge the current competitiveness of the proposed method.

**Limitations:**

Yes

**Strengths And Weaknesses:**

Strengths

1. The paper studies an important practical issue in GFlowNet-based molecular generation, namely the effect of trajectory truncation when the chemical space is extremely large. The motivation of separating valid molecules from truncated fragments is intuitive and relevant to molecular optimization tasks.

2. The proposed architecture is relatively simple and can be integrated into existing GFlowNet frameworks with minimal modifications.

3. The paper provides both theoretical intuition and empirical evaluations across multiple benchmarks, including single-objective, pocket-conditioned, and multi-objective molecular generation tasks.

Weaknesses

As detailed in Questions.

---

> ### Author Rebuttal · Authors · 2026-03-31
>
> We thank the reviewer for the detailed feedback. We address each concern below.
>
> **Q1. Lack of convincing evidence for the core phenomenon.**
>
> We provide direct evidence that truncation causes mode collapse in standard GFlowNets. We tracked [Uniqueness over training iterations](https://anonymous.4open.science/r/LeakGFN-464A/figures/uniqueness_over_iterations.png) across all five tasks, revealing a clear task-dependent pattern:
>
> - **JNK3, GSK3$\\beta$**: GFN-FM's Uniqueness collapses during training with extreme variance (JNK3: 0.444±0.468, Appendix B), indicating the model repeatedly generates the same molecules. This is the signature of truncation distorting the learned policy: flow conservation over the truncated DAG causes over-concentration on remaining paths. LeakGFN maintains stable Uniqueness (0.930±0.063).
> - **DRD2**: Both methods collapse early, but from a narrow reward landscape, not truncation. Target molecules mostly complete within $L_{max}=8$.
> - **QED, SA**: Both maintain high Uniqueness, consistent with negligible truncation for these tasks.
>
> This demonstrates that truncation-induced collapse is not hypothetical: it is the dominant failure mode on tasks requiring larger molecules.
>
> The reviewer notes that $L_{max}$ is often chosen large enough. However, Table 3 shows that increasing $L_{max}$ makes the problem *worse* for standard methods (GFN-DB: 0.934→0.096 from $L_{max}$=8→12), because forced terminals grow combinatorially. LeakGFN degrades gracefully (0.797→0.435), maintaining 4× higher performance. In practice, the optimal $L_{max}$ is rarely known a priori, making this robustness essential.
>
> **Q2. Unclear definition of "flow leakage".**
>
> We apologize for the ambiguity. We formally define:
>
> **Flow leakage**: the probability mass allocated to forced terminal states $\\mathcal{T}\_{\\text{forced}}$ under the learned distribution. Standard GFlowNets learn $P(x) \\propto R(x)$ for $x \\in \\mathcal{T}\_{\\text{valid}} \\cup \\mathcal{T}\_{\\text{forced}}$, but the correct target (Eq. 4) is defined only over $\\mathcal{T}\_{\\text{acc}}$. The leaked mass is $\\sum\_{x \\in \\mathcal{T}\_{\\text{forced}}} P(x)$. The severity varies by task: as shown in Q1, tasks requiring larger molecules (JNK3, GSK3$\\beta$) suffer severe Uniqueness collapse, while tasks with small molecules (QED, SA) are unaffected.
>
> Beyond probability mass, flow leakage also distorts the *policy*: truncation cuts paths in the DAG that exist in the original chemical space, and flow conservation enforced over this truncated DAG causes the model to over-concentrate on remaining paths, leading to mode collapse. This is why the effect is more severe than the raw probability mass suggests.
>
> **Q3. Triviality of Theorem 3.1.**
>
> We respectfully disagree that Theorem 3.1 is a direct consequence of the standard GFlowNet framework. The distinction is subtle but important:
>
> The standard correctness theorem guarantees $P(x) \\propto R(x)$ over all terminal states. Under truncation, this means $P(x) \\propto R(x)$ for $x \\in \\mathcal{T}\_{\\text{valid}} \\cup \\mathcal{T}\_{\\text{forced}}$, which is precisely the wrong distribution. The standard framework faithfully recovers a distribution that includes forced terminals, and this is the core problem our paper identifies.
>
> Theorem 3.1 operates on $F\_{valid}$, which is not a standard GFlowNet flow. It is a decomposed flow $F\_{chem} \\cdot \\lambda$ with a non-standard terminal set (forced terminals are treated as non-terminals for $F\_{valid}$). Applying the standard theorem to $F\_{valid}$ is not immediate because: (i) $F\_{valid}$ is defined through a multiplicative decomposition not present in standard GFlowNets, (ii) the terminal set differs between $F\_{valid}$ and $F\_{chem}$, and (iii) the asymmetric training objective (Table 1) creates a non-standard learning setup. Theorem 3.1 verifies that despite these modifications, the correct distribution over $\\mathcal{T}\_{\\text{acc}}$ is recovered.
>
> That said, we agree the proof technique is straightforward once the setup is established, and the primary contribution is architectural and empirical. We will clarify this positioning in the revision.
>
> **Q4. Limited comparison with recent methods.**
>
> Our main experiments (Table 2) deliberately use standard GFlowNet objectives (FM, TB, SubTB, DB) on the same architecture, vocabulary, and oracle to provide a **controlled comparison** that isolates the effect of our dual-head decomposition.
>
> Beyond this controlled setting, we also evaluate in more complex scenarios where comparison with other paradigms is natural. For pocket-conditioned generation (Table 5), LeakGFN+TacoGFN outperforms recent diffusion-based methods (TargetDiff, DecompDiff) and autoregressive methods (Pocket2Mol) on binding affinity (Vina Dock: $-8.61$ vs $-8.35$). For multi-objective optimization (Table 6), integration with HN-GFN achieves 22% higher hypervolume than prior methods.

---

> > ### Author Rebuttal · Reviewer_Zutc · 2026-04-02
> >
> > Thanks for your responses. I belive including the updated definitions and experiments in the revision will make the manuscript stronger.

---

> > > ### Author Response · Authors · 2026-04-03
> > >
> > > We sincerely thank the reviewer for the constructive feedback and for acknowledging our responses. As suggested, we will include the clarified definition of flow leakage and the training dynamics analysis in the revised manuscript to strengthen the presentation. Thank you again for the valuable suggestions.

---

### Official Review · Reviewer_yniq · 2026-03-12

**Soundness:** 4
**Presentation:** 4
**Significance:** 3
**Originality:** 3
**Overall Recommendation:** 5
**Confidence:** 4

**Summary:**

The paper's core claim is that when one imposes a maximum trajectory length in GFNs, they incorrectly treat boundary-hit partial constructions as terminal states, thereby assigning probability mass to incomplete molecular fragments rather than only to valid accessible molecules. The proposed solution, LeakGFN, introduces a dual-head decomposition: a chemical-flow head $F_{chem}$ over the full chemical space and a valid-flow head $F_{valid} =F_{chem}⋅λ$, where $λ$ is meant to capture whether flow through a state-action pair still reaches an accessible valid terminal under truncation. The paper provides a conditional correctness theorem for sampling from $F_{valid}$ and reports strong empirical results on five molecular optimization tasks, plus integrations into pocket-conditioned and multi-objective settings.

**Compliance With Llm Reviewing Policy:**

Affirmed.

**Final Justification:**

The authors offered detailed explanations and full clarifications for my questions/raised weaknesses during rebuttal. I maintain my original assessment and recommendation for acceptance.

**Key Questions For Authors:**

- How sensitive is the method to the quality and calibration of oracle rewards on forced-terminal partial molecules? Have you checked cases where partial states receive unexpectedly high surrogate scores?
- Would it be feasible for the authors to provide some measures of how much sampled probability mass lands on forced terminals under a standard GFN? Or report the fraction of trajectories ending in forced terminals

**Limitations:**

yes

**Strengths And Weaknesses:**

**Strengths**
- Well-motivated problem formulation: the paper identifies a real and under-discussed mismatch between the intended GFN target distribution and what is actually learned. The distinction between valid terminals and forced terminals is clearly stated, and the newly introduced objective is formalized cleanly in Eqs. 15-17.
- Apart from the introduced formalization, the paper clearly states differences to epsilon-exploration and samples actions proportionally to the chemical flow. This results in eliminating the need for explicit action masking.
- The paper also nicely outlines theoretical guarantees for recovering correct distributions. The theorem is simple, but it meaningfully connects the algorithmic construction to the desired probabilistic sampling.
- The paper extensively evaluates against GFNs with various objective functions and other strong molecule generation baselines. The robustness-to-$L_{max}$ experiment is compelling: on DRD2, trajectory-level methods collapse badly when $L_{max}$ increases, while LeakGFN degrades much more gracefully. That pattern aligns with the paper’s motivating claim about truncation leakage rather than looking like an arbitrary benchmark win.
- Additionally, the paper demonstrates empirical wins on SBDD tasks and MOO.

**Weaknesses**
- The paper argues that in fragment-based generation every state is still a valid molecular graph evaluable by the oracle and that such states usually receive low rewards. But this assumption may depend strongly on the benchmark oracle. For learned activity surrogates on molecular graphs, an incomplete-yet-chemically-valid fragment graph may receive arbitrary or even high scores for reasons unrelated to “reachability to a good completed molecule.” A discussion of these matters would be encouraged.
- The ablation removes exploration and removes the valid head, which is useful, but it still does not fully disentangle whether the main benefit comes from the decomposition itself, the self-loop stop calibration, the exact training objective at forced terminals, or the exploration policy

---

> ### Author Rebuttal · Authors · 2026-03-31
>
> We thank the reviewer for the constructive feedback and engagement. We address weaknesses and key questions together below.
>
> ### Weaknesses
>
> **Weakness 1: Oracle dependence.** This is a fair point. We agree that learned surrogate oracles may assign arbitrary scores to incomplete fragments, which could reduce the practical severity of flow leakage in some settings. Our experiments use well-calibrated oracles (random forest on bioactivity data for kinases, RDKit QED/SA) where incomplete fragments consistently receive low rewards due to missing pharmacophoric features.
>
> Importantly, even if some fragments receive high rewards, LeakGFN remains robust because $\\lambda$ learns reachability from the *structural* asymmetry in training (Table 1), not from reward magnitude. At forced terminals, the training drives $\\sum_a F_{valid}(s,a) \\to 0$ while $F_{chem}(s,a) > 0$, pushing $\\lambda(s,a)$ toward 0 **regardless of R(s)**. So even with a miscalibrated oracle assigning high scores to fragments, $\\lambda$ still down-weights forced terminals in sampling.
>
> **Weakness 2: Ablation granularity.** Table 4 provides a comparison that isolates each contribution:
>
> | Setting | HM (JNK3) |
> |---|---|
> | Single-head + $\\epsilon$-exploration (GFN-FM) | 0.403 |
> | Dual-head + $\\epsilon$-exploration (w/o $\\pi_{explore}$) | 0.62 |
> | Dual-head + flow-guided (LeakGFN full) | 0.64 |
>
> The dual-head decomposition provides the dominant improvement (+0.22), while flow-guided exploration contributes modestly (+0.02). Regarding self-loop calibration: this ensures $F(s, \\texttt{stop}) = R(s)$ at terminals, which is a *necessary condition* for the asymmetric terminal treatment to work (Section 3.4). Without it, the constraint $\\sum_a F_{valid}(s,a) = 0$ at forced terminals cannot be enforced. It is therefore an integral component rather than an independent source of improvement.
>
> ### Key Questions
>
> **Q: Oracle robustness: what if partial states receive high surrogate scores?**
>
> As discussed in Weakness 1, $\\lambda$ at forced terminals is driven toward 0 by structural training asymmetry, independent of R(s). A high reward is captured by $F(s, \\texttt{stop})$, but sampling uses $F_{valid}$ which down-weights these states. One caveat: if a high-reward molecule is *only* reachable via forced termination, LeakGFN will not sample it. This indicates $L_{max}$ should be increased, and Table 3 shows LeakGFN degrades gracefully as $L_{max}$ grows.
>
> **Q: Fraction of probability mass on forced terminals / forced terminal statistics?**
>
> Directly measuring the probability mass on forced terminals requires distinguishing voluntary stops at $L_{max}$ from forced stops, which is nontrivial in practice since both occur at the same trajectory depth. Instead, we provide the downstream consequence of truncation, which is more directly informative: **mode collapse during training**.
>
> We tracked [Uniqueness over training iterations](https://anonymous.4open.science/r/LeakGFN-464A/figures/uniqueness_over_iterations.png) across all five tasks. The results reveal a clear task-dependent pattern:
>
> - **JNK3, GSK3$\\beta$**: GFN-FM's Uniqueness collapses during training with extreme variance (JNK3: 0.444±0.468, Appendix B), indicating the model repeatedly generates the same molecules. This is the signature of truncation distorting the policy. LeakGFN maintains stable Uniqueness (0.930±0.063).
> - **DRD2**: Both methods collapse early, but from a narrow reward landscape, not truncation. This explains why LeakGFN shows limited improvement at $L_{max}=8$ but becomes decisive at $L_{max} \\geq 10$ (Table 3).
> - **QED, SA**: Both maintain high Uniqueness throughout, consistent with the small molecule sizes required for these tasks.
>
> This pattern directly demonstrates that truncation-induced mode collapse is concentrated on tasks requiring larger molecules, and that LeakGFN specifically resolves it.

---

> > ### Author Rebuttal · Reviewer_yniq · 2026-04-02
> >
> > Thank you for the replies and clarifications. My concerns / questions have been fully addressed, and I maintain my positive score in favour of acceptance.

---

> > > ### Author Response · Authors · 2026-04-03
> > >
> > > We thank the reviewer for the positive evaluation and for confirming that our responses fully addressed the raised concerns. Thank you again for the constructive engagement throughout the review process.

---

### Official Review · Reviewer_R74h · 2026-03-13

**Soundness:** 2
**Presentation:** 3
**Significance:** 3
**Originality:** 2
**Overall Recommendation:** 4
**Confidence:** 4

**Summary:**

The paper proposes LeakGFN, a variant of GFlowNets (GFN) designed to address the issue of truncation leading to probability mass distortion. In particular, as GFNs typically truncate generated sequences to render computation tractable, they end up assigning forced probability mass to forced terminations (often incomplete molecules).

To resolve the issue, LeakGFN learns two heads, a valid head and a chemical head with the valid head learning the amount of flow that can reach accessible terminal states while the chemical head learns the unrestricted flow over the full state space. By sampling from the valid head, they show they can recover the correct distribution.

Experimentally, they show LeakGFN outperforms other methods in terms of the harmonic mean of mean score, diversity and uniqueness. They also show their method can be used in conjunction with TacoGFN to further improve performance.

**Compliance With Llm Reviewing Policy:**

Affirmed.

**Final Justification:**

I recommend acceptance of the paper as the rebuttal has addressed my main concern by better explaining the benefits of the method lie in improving the training dynamics with respect to forced termination (as supported by their evidence).

These points should be made more explicit in an updated version of the paper. In addition, more experiments illustrating the issue of forced termination in terms of loss of gradient signal and "flow conservation forc[ing] the remaining paths to absorb all the flow, causing over-concentration and mode collapse" would strengthen their explanation.

**Key Questions For Authors:**

- I'm a bit surprised the FM style loss works so well as it was my understanding that most GFN papers use TB/SubTB now. Is it necessary to be able to use the valid/chemical heads? Or was there some other reason its used?
- In App. D, it feels like the learned valid fraction lambda should be noticeably lower towards the last step (instead of 0.1 lower). Are the learned values expected?
- Do you expect any other generalization benefits from this loss setup?
- In Eq. 10, how are the $F_{chem}(s,a)$ learned at forced terminals if the learning stops at truncation?

**Limitations:**

Not discussed.

**Strengths And Weaknesses:**

**Strengths**
- The paper is well-structured and easy to follow
- The idea and setup are novel, learning both a chemical and valid flow head is an interesting way to decompose flow.
- Extensive experiments show the benefits of the method in a variety of tasks (as well as its potential to be used in conjunction with other methods).
- The paper ablates the different design choices (including the exploration policy).

**Weaknesses**
- I am not fully convinced how forced terminal states corresponding to incomplete structures is any different than terminations earlier in the graph that yield incomplete structures.
  - Under an ideal reward function, these should be assigned 0 reward and thus 0 probability mass.
  - If they are assigned any probability mass it must be that they have positive reward in which case its an issue of the reward function, not necessarily the truncation.
  - Would just extending L max by 1 and setting the flow at truncated states to 0 yield the same distribution over states?

I am willing to increase my score if this concern is addressed/clarified.

---

> ### Author Rebuttal · Authors · 2026-03-31
>
> We thank the reviewer for the thoughtful questions and engagement. We address the weaknesses and key questions together, as they are closely related.
>
> ### Weakness
>
> These are important concerns that touch on our core design motivation. We believe the central point hinges on one key distinction: **voluntary vs. forced termination**. In fragment-based generation, a model can stop at any step via the stop action. If it stops early, that is the agent's *decision*, and the resulting molecule enters the distribution as an intentional output. Forced terminals are different: the agent would have continued, but the environment ($L_{max}$) prevents it. GFlowNet's flow conservation assumes termination is the agent's choice; forced termination violates this premise, creating a distribution over $\\mathcal{T}\_{valid} \\cup \\mathcal{T}\_{forced}$ rather than the intended $\\mathcal{T}\_{valid}$ alone.
>
> Setting R(x)=0 at forced terminals seems like a natural fix, but it fails for two reasons. First, **gradient vanishing**: the model pushes $F_{in}(x) \\to 0$, after which these trajectories contribute near-zero gradients to the shared encoder (analogous to dying ReLU). Excluding them from the loss wastes training samples. This creates a dilemma: oracle reward R(s)>0 preserves gradients but pollutes the target support, while R(x)=0 corrects the support but starves the model. Second, and more fundamentally, R(x)=0 only tells the model to *avoid* forced terminals. It provides no mechanism to prevent flow distortion near the truncation boundary. When truncation cuts outgoing paths, flow conservation forces the remaining paths to absorb all the flow, causing over-concentration and mode collapse.
>
> This is why the problem is about **truncation, not reward**. Even with a perfect reward function, truncation cuts paths in the original chemical DAG. The model cannot explore continuations that exist in the full chemical space, and the resulting policy distortion is independent of reward accuracy. On JNK3, this manifests as GFN-FM's Uniqueness collapsing to 0.444±0.468 (Appendix B, see also [training dynamics](https://anonymous.4open.science/r/LeakGFN-464A/figures/uniqueness_over_iterations.png)); LeakGFN recovers this to 0.930±0.063.
>
> Regarding extending $L_{max}$ by 1 with flow=0: this can be viewed as a **special case** of LeakGFN, equivalent to hard-setting $\\lambda=0$ at a single fixed depth while keeping $\\lambda=1$ elsewhere. LeakGFN generalizes this by learning $\\lambda$ softly across all depths. The hard boundary creates a sharp discontinuity that can destabilize learning; moreover, extending $L_{max}$ merely shifts the boundary without resolving the structural issue.
>
> LeakGFN resolves the dilemma by decomposing flow: $F_{chem}$ maintains positive outgoing flow at forced terminals, preventing flow distortion near the boundary and preserving gradients, while $\\lambda$ down-weights forced terminals in sampling. Our $\\alpha=0$ analysis (Appendix C) confirms: even without explicit chemical flow supervision, the decomposition alone improves over GFN-FM (0.591 vs 0.356 at $L_{max}=10$ on DRD2).
>
> ### Key Questions
>
> **Q1. Why FM?** LeakGFN decomposes flow at the *state level*, and FM provides state-level supervision, making it a natural pairing. The architecture is objective-agnostic: TacoGFN (Table 5) uses TB internally, and our integration improved Vina Dock from $-8.24 \\to -8.61$.
>
> **Q2. The $\\lambda$ decrease seems modest.** Figure 2 averages over the 20K-30K training range, which dilutes the signal by including earlier, less-converged checkpoints. At the final checkpoint (iter 30,000), the decrease is much clearer: **0.95 at initial step → 0.71 at max step** (Δ=0.24), as shown in [this figure](https://anonymous.4open.science/r/LeakGFN-464A/figures/reb.png). The steepest drop occurs at steps 1→2 (0.94→0.81), aligning with scaffold determination in fragment-based generation. The analysis uses fixed pre-sampled length-8 trajectories, so this reflects learned reachability rather than sampling bias.
>
> **Q3. Generalization benefits?** We demonstrate plug-and-play integration into TacoGFN (pocket-conditioned, Table 5) and HN-GFN (multi-objective, Table 6). The core principle, separating "what to reward" from "what is reachable," applies to any GFlowNet with truncation constraints. We also observe consistent variance reduction (GSK3β: std 0.071→0.008), suggesting implicit regularization.
>
> **Q4. How is $F_{chem}$ learned at forced terminals?** At forced terminals, $F_{chem}$ is treated as non-terminal (Table 1): only flow conservation is applied, maintaining $F_{chem}(s,a)>0$. Meanwhile, $F_{valid}$ treats them as terminals, forcing $\\sum_a F_{valid}(s,a) \\to 0$. Since $F_{valid}=F_{chem}\\cdot\\lambda$ with $F_{valid}\\to 0$ and $F_{chem}>0$, at convergence $\\lambda(s,a)$ is driven toward 0. This learning signal propagates backward through the DAG via flow matching.

---

> > ### Author Rebuttal · Reviewer_R74h · 2026-04-01
> >
> > I thank the authors for their thorough response.
> >
> > The explanation clarifies the benefits of the method to me, the main advantage comes not from changing the optimal learned policy but from likely significant issues in the training dynamics of the forced termination. I would suggest some version of your explanation is described more explicitly in the paper.
> >
> > My other concerns have been addressed and I will raise my score accordingly.

---

> > > ### Author Response · Authors · 2026-04-02
> > >
> > > We sincerely thank for the thoughtful reconsideration and for acknowledging our response.
> > >
> > > We appreciate the suggestion to make the training dynamics perspective more explicit in the paper. We will incorporate a clearer discussion of this aspect in the revised version.
> > >
> > > Thank you again for the constructive feedback.

---

### Decision · Program_Chairs · 2026-04-30

**Decision:**

Accept (regular)

**Comment:**

The paper proposes LeakGFN, a dual-head GFlowNet architecture that addresses flow leakage caused by trajectory truncation in molecular generation by decomposing flow into chemical and valid components. Three of four reviewers recommend acceptance (scores 4, 5, 5), with all marking concerns fully resolved.

The main contribution (that the dual-head decomposition improves training dynamics near the truncation boundary rather than merely changing the target distribution) was well-supported by ablations showing the decomposition accounts for the dominant improvement (+0.22 HM) independent of exploration strategy.

The authors also provided the R(x)=0 ablation requested by Reviewer 4DnB, demonstrating it degrades performance in 3/4 conditions, which strengthens the case for the decomposition approach.

The paper should make the training dynamics argument and formal definition of flow leakage more prominent in the revision.